# Adversarial Attacks on Deep Graph Matching

**Zijie Zhang**
Auburn University
zzz0092@auburn.edu

**Zeru Zhang**
Auburn University
zzz0054@auburn.edu

**Yang Zhou**
Auburn University
yangzhou@auburn.edu

**Yelong Shen**
Microsoft Dynamics 365 AI
yeshe@microsoft.com

**Ruoming Jin**
Kent State University
rjin1@kent.edu

**Dejing Dou**
University of Oregon, Baidu Research
dou@cs.uoregon.edu, doudejing@baidu.com

## Abstract

Despite achieving remarkable performance, deep graph learning models, such as node classification and network embedding, suffer from harassment caused by small adversarial perturbations. However, the vulnerability analysis of graph matching under adversarial attacks has not been fully investigated yet. This paper proposes an adversarial attack model with two novel attack techniques to perturb the graph structure and degrade the quality of deep graph matching: (1) a kernel density estimation approach is utilized to estimate and maximize node densities to derive imperceptible perturbations, by pushing attacked nodes to dense regions in two graphs, such that they are indistinguishable from many neighbors; and (2) a meta learning-based projected gradient descent method is developed to well choose attack starting points and to improve the search performance for producing effective perturbations. We evaluate the effectiveness of the attack model on real datasets and validate that the attacks can be transferable to other graph learning models.

## 1   Introduction

Graph matching is one of the most important research topics in the graph domain, which aims to match the same entities (i.e., nodes) across two or more graphs [91, 98, 43, 46, 48, 72, 54, 105, 13, 75]. It has been widely applied to many real-world applications ranging from protein network matching in bioinformatics [33, 63], user account linking in different social networks [62, 51, 100, 37, 101, 21, 38], and knowledge translation in multilingual knowledge bases [87, 124], to geometric keypoint matching in computer vision [22]. Existing research efforts on graph matching can be classified into three broad categories: (1) structure-based techniques, which rely only upon the topological information to match two or multiple input graphs [43, 49, 95, 46, 54, 105, 13, 96, 84, 40, 67, 57, 38, 24]; (2) attribute-based approaches, which utilize highly discriminative structure and/or attribute features for ensuring the matching effectiveness [93, 94, 51, 10, 65, 16, 88, 100, 28, 39, 18, 97, 50, 52, 22]; and (3) heterogeneous methods, which employ heterogeneous structural, content, spatial, and temporal features to further improve the matching performance [92, 34, 44, 98, 83, 99, 77, 59, 102, 103, 21].

Recent literature has shown that both traditional and deep graph learning algorithms remain highly sensitive to adversarial attacks, i.e., carefully designed small perturbations in graph structure and attributes can cause the models to produce wrong prediction results [14, 126, 64, 125, 123, 69, 90, 74, 45, 85, 80, 127]. We have witnessed various effective attack models to cause failures of

node classification [14, 126, 74, 86, 125, 71, 19, 70], community detection [9, 78, 7, 41], network embedding [6, 4, 5], link prediction [104], similarity search [15], malware detection [30], and knowledge graph embedding [90]. However, there is still a paucity of analyses of the vulnerability of graph matching under adversarial attacks, which is much more difficult to study. Most of the existing models to fool other graph learning tasks conduct the adversarial attacks on a single graph but the graph matching task analyzes both intra-graph and inter-graph interactions of multiple graphs. In this work, we aim to answer the following questions: (1) Are graph matching algorithms sensitive to small perturbation of graph structure? (2) How do we develop effective and imperceptible perturbations for degrading the performance of deep graph matching models?

A large number of research advances in adversarial attacks on graph data utilize iterative gradient-based methods to produce effective adversarial perturbations that fool a graph learning model [68, 76, 14, 125, 86, 71, 89, 8]. However, a recent study reports that the iterative gradient-based methods, such as Fast Gradient Sign Method (FGSM) [26] and Projected Gradient Descent (PGD) [47], start the attacks from original examples and add perturbations monotonically along the direction of gradient descent, resulting in a lack of diversity and adaptability of generated iterative trajectories [61]. This often leads to invalid attacks since the iterative trajectories have difficulties crossing decision boundary of target learning model with small perturbation. Can we find a shortcut across the decision boundary to derive more effective attacks by beginning from good attack starting points in the graph matching?

Traditionally, graph matching techniques are based on the assumption of feature consistency across graphs: Two nodes in different graphs are more likely to be found to be matching if they have similar topological and/or attribute features in respective graphs [98, 93, 10, 17, 28, 96]. These methods compute the similarity (or distance) scores between pairwise nodes across graphs and choose the node pairs with largest similarity (or smallest distance) as matching results [101, 88, 39, 38]. Intuitively, if an attacker perturbs a node by throwing it into a dense region in the graph with many similar nodes, i.e., a pile of nodes similar to each other, such that this attacked node is similar to many neighbors, then it is hard for humans or defender programs to recognize it from the node pile. In addition, if two matched nodes are simultaneously moved to such dense regions in respective graphs, then this dramatically increases the difficulty in matching them correctly among many similar candidate nodes.

To our best knowledge, this work is the first to study adversarial attacks on graph matching.

We propose to utilize kernel density estimation (KDE) technique to estimate the probability density function of nodes in two graphs, to understand the intrinsic distribution of graphs. By maximizing the estimated densities of nodes to be attacked, we push them to dense regions in respective graphs to generate adversarial nodes that are indistinguishable from many neighbors in dense regions. This increases the chance of producing wrong matching results as well as reduces the risk of perturbations being detected by humans or by defender programs. Our analysis is the first to introduce the KDE technique to conduct imperceptible attacks on graph data.

Searching for good attack starting points in large graphs is computationally inefficient. We develop a meta learning-based projected gradient descent (MLPGD) model to quickly adapt to a variety of new search tasks on multiple batches of target nodes for deriving effective attacks. However, the MLPGD model is non-smooth and non-differential, as the perturbation is a multi-step process and the projection at each step is non-differential. A Gaussian smoothing method is designed to approximate a smoothed model, and a Monte Carlo REINFORCE method is used to estimate the model gradient.

Empirical evaluation on real datasets demonstrates the superior performance of the GMA model against several state-of-the-art adversarial attack methods on graph data. Moreover, we validate that the attack strategies are transferable to other popular graph learning models in Appendix A.2.

## 2   Problem Definition

Given two graphs $G^1$ and $G^2$ to be matched, each is denoted as $G^s = (V^s, E^s)$ ($s = 1$ or $2$), where $V^s = \{v_1^s, \cdots, v_{N^s}^s\}$ is the set of $N^s$ nodes and $E^s = \{(v_i^s, v_j^s) : 1 \leq i, j \leq N^s\}$ is the set of edges. Each $G^s$ has an $N^s \times N^s$ binary adjacency matrix $\mathbf{A}^s$, where each entry $\mathbf{A}_{ij}^s = 1$ if there exists an edge $(v_i^s, v_j^s) \in E^s$; otherwise $\mathbf{A}_{ij}^s = 0$. $\mathbf{A}_{i:}^s$ specifies the $i^{th}$ row vector of $\mathbf{A}^s$. In this paper, if there are no specific descriptions, we use $\mathbf{v}_i^s$ to denote a node $v_i^s$ itself and its representation $\mathbf{A}_{i:}^s$, i.e., $\mathbf{v}_i^s = \mathbf{A}_{i:}^s$ and we utilize $\mathbf{v}_{ij}^s$ to specify the $j^{th}$ dimension of $\mathbf{v}_i^s$, i.e., $\mathbf{v}_{ij}^s = \mathbf{A}_{ij}^s$.

The dataset is divided into two disjoint sets $D'$ and $D$. The former denotes a set of known matched node pairs $D' = \{(\mathbf{v}_i^1, \mathbf{v}_k^2)|\mathbf{v}_i^1 \leftrightarrow \mathbf{v}_k^2, \mathbf{v}_i^1 \in V^1, \mathbf{v}_k^2 \in V^2\}$, where $\mathbf{v}_i^1 \leftrightarrow \mathbf{v}_k^2$ indicates that two nodes $\mathbf{v}_i^1$ and $\mathbf{v}_k^2$ belong to the same entity. The latter, denoted by $D = \{(\mathbf{v}_i^1, \mathbf{v}_k^2)|\mathbf{v}_i^1 \leftrightarrow \mathbf{v}_k^2, \mathbf{v}_i^1 \in V^1, \mathbf{v}_k^2 \in V^2\}$, is used to evaluate the graph matching performance, where the nodes (but not their matchings) are also observed during training. The goal of graph matching is to utilize $D'$ as the training data to identify the one-to-one matching relationships between nodes $\mathbf{v}_i^1$ and $\mathbf{v}_k^2$ in the test data $D$. By following the same idea in existing efforts [101, 88, 39, 38], this paper aims to minimize the distances between projected source nodes $M(\mathbf{v}_i^1) \in D'$ and target ones $\mathbf{v}_k^2 \in D'$. The node pairs $(\mathbf{v}_i^1, \mathbf{v}_k^2) \in D$ with the smallest distances are selected as the matching results.

$$\min_M L \text{ where } L = \mathbb{E}_{(\mathbf{v}_i^1, \mathbf{v}_k^2) \in D'} \|M(\mathbf{v}_i^1) - \mathbf{v}_k^2\|_2^2 \tag{1}$$

where $M$ denotes an injective one-to-one matching function $M : \mathbf{v}_i^1 \in V^1 \mapsto \mathbf{v}_k^2 \in V^2$.

The adversarial attack problem is defined as maximally degrading the matching performance of $M$ on the test data $D$ by injecting edge perturbations (including edge insertion and deletion) into $G^s = (V^s, E^s)$ ($s = 1$ or $2$), leading to two adversarial graphs $\hat{G}^s = (\hat{V}^s, \hat{E}^s)$. We assume the attacker has limited capability, so that he/she can only make small perturbations.

## 3  Imperceptible Attacks with Node Density Estimation and Maximization

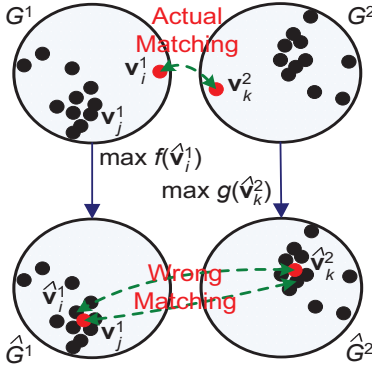

Figure 1: Imperceptible Attacks

Intuitively, in Eq.(1), if there exist nodes $\mathbf{v}_j^1$ similar to $\mathbf{v}_i^1$, i.e., $\mathbf{v}_j^1 \approx \mathbf{v}_i^1$, such that $\|M(\mathbf{v}_j^1) - \mathbf{v}_k^2\|_2^2 < \|M(\mathbf{v}_i^1) - \mathbf{v}_k^2\|_2^2$, then a wrong matching $(\mathbf{v}_j^1, \mathbf{v}_k^2)$ will be generated. In addition, if there are many such $\mathbf{v}_j^1$s around $\mathbf{v}_i^1$, then it is hard to recognize $\mathbf{v}_i^1$ from a pile of similar nodes. Thus, if we move $\mathbf{v}_i^1$ to dense regions that contain many similar $\mathbf{v}_j^1$s, then this dramatically increases the possibility of deriving the wrong matching $(\mathbf{v}_j^1, \mathbf{v}_k^2)$ among many similar candidate nodes. Also, as many $\mathbf{v}_j^1$s are around the adversarial node $\hat{\mathbf{v}}_i^1$, it is difficult for humans or defender programs to detect $\hat{\mathbf{v}}_i^1$, as shown in a toy example in Figure 1. Motivated by this, we propose to employ kernel density estimation (KDE) method to generate imperceptible perturbations. In statistics, the KDE is to estimate the probability density function $f(x)$ of a random variable $x$ with unknown distribution [55]. It helps reveal the intrinsic distribution.

Concretely, let $\mathbf{v}^1$ be a $N^1$-dimensional random variable to denote all nodes $\{\mathbf{v}_i^1, \cdots, \mathbf{v}_{N^1}^1\}$ in graph $G^1$ with an unknown density $f$. A function $\hat{f}(x)$ is estimated to best approximate $f(x)$.

$$\hat{f}(\mathbf{v}^1) = \frac{1}{N^1 \det(\mathbf{B})} \sum_{i=1}^{N^1} \mathcal{K}\left(\mathbf{B}^{-1}\left(\mathbf{v}^1 - \mathbf{v}_i^1\right)\right) \tag{2}$$

where $\det(\cdot)$ denotes the determinant operation. $\mathbf{B} > 0$ is a bandwidth to be estimated. It is an $N^1 \times N^1$ diagonal matrix $\mathbf{B} = diag(b_1, \cdots, b_{N^1})$, which has strong influence on the density estimation $\hat{f}(\mathbf{v}^1)$. A good $\mathbf{B}$ should be as small as the data can allow. $\mathcal{K}$ is a product symmetric kernel that satisfies $\int \mathcal{K}(x)dx = 1$ and $\int x\mathcal{K}(x)dx = 0$. The above vector-wise form $\hat{f}(\mathbf{v}^1)$ can be rewritten as an element-wise form, where $\mathbf{v}_j^1$ represents the $j^{th}$ dimension in $\mathbf{v}^1$.

$$\hat{f}(\mathbf{v}^1) = \frac{1}{N^1} \sum_{i=1}^{N^1} \prod_{j=1}^{N^1} \frac{1}{b_j} \mathcal{K}\left(\frac{\mathbf{v}_j^1 - \mathbf{v}_{ij}^1}{b_j}\right) \tag{3}$$

The derivative $\frac{\partial \hat{f}(\mathbf{v}^1)}{\partial b_j}$ w.r.t. each bandwidth $b_j$ in $\mathbf{B}$ is computed as follows, where $K(x) = \frac{d \log \mathcal{K}(x)}{dx}$.

$$\frac{\partial \hat{f}(\mathbf{v}^1)}{\partial b_j} = \frac{1}{N^1} \sum_{i=1}^{N^1} \frac{\partial \left[\prod_{l=1}^{N^1} \frac{1}{b_l} \mathcal{K}\left(\frac{\mathbf{v}_l^1 - \mathbf{v}_{il}^1}{b_l}\right)\right]}{\partial b_j} = -\frac{1}{N^1} \sum_{i=1}^{N^1} \left(\frac{1}{b_j} + \frac{\mathbf{v}_l^1 - \mathbf{v}_{il}^1}{b_j^2} K\left(\frac{\mathbf{v}_l^1 - \mathbf{v}_{il}^1}{b_j}\right)\right) \prod_{l=1}^{N^1} \frac{1}{b_l} \mathcal{K}\left(\frac{\mathbf{v}_l^1 - \mathbf{v}_{il}^1}{b_l}\right) \tag{4}$$

Traditional KDE methods often fail on high-dimensional data [29, 60, 32, 36], when bandwidths need to be selected for each dimension. A greedy search method is utilized to select bandwidths in

the KDE: If a dimension $j$ is insignificant, then changing the bandwidth $b_j$ for that dimension should have a weak impact on $\hat{f}(\mathbf{v}^1)$, while the changing $b_j$ for an important $j$ should cause a large change in $\hat{f}(\mathbf{v}^1)$. Fortunately, $\frac{\partial \hat{f}(\mathbf{v}^1)}{\partial b_j}$ can differentiate these two types of dimensions. Based on the above analysis, we greedily decrease $b_j$ with a sequence $b_0, b_0 s, b_0 s^2, \cdots$ for a parameter $0 < s < 1$, until $b_j$ is smaller than a certain threshold $\tau_j$, to see if a small change in $b_j$ can result in a large change in $\hat{f}(\mathbf{v}^1)$. The method also offers a good way to estimate $\big[\frac{\partial \hat{f}(\mathbf{v}^1)}{\partial b_1}, \cdots, \frac{\partial \hat{f}(\mathbf{v}^1)}{\partial b_{N^1}}\big]$ along a sparse path.

Concretely, $\hat{f}(\mathbf{v}^1)$ is estimated by beginning with an initial $\mathbf{B} = diag(b_0, \cdots, b_0)$ for a large $b_0$, and then estimate $\frac{\partial \hat{f}(\mathbf{v}^1)}{\partial b_j}$ as follows and decrease $b_j$ if $\frac{\partial \hat{f}(\mathbf{v}^1)}{\partial b_j}$ is large.

$$\frac{\partial \hat{f}(\mathbf{v}^1)}{\partial b_j} = \frac{1}{N^1} \sum_{i=1}^{N^1} \frac{\partial\big[\prod_{l=1}^{N^1} \frac{1}{b_l} \mathcal{K}(\frac{\mathbf{v}_l^1 - \mathbf{v}_{il}^1}{b_l})\big]}{\partial b_j} = \frac{1}{N^1} \sum_{i=1}^{N^1} \frac{\mathcal{K}(\frac{\mathbf{v}_j^1 - \mathbf{v}_{ij}^1}{b_j})}{\mathcal{K}(\frac{\mathbf{v}_j^1 - \mathbf{v}_{ij}^1}{b_j})} \prod_{l=1}^{N^1} \mathcal{K}(\frac{\mathbf{v}_l^1 - \mathbf{v}_{il}^1}{b_l}) = \frac{1}{N^1} \sum_{i=1}^{N^1} \frac{\partial \hat{f}(\mathbf{v}_i^1)}{\partial b_j}$$

(5)

The corresponding variance $\mathrm{Var}\Big(\frac{\partial \hat{f}(\mathbf{v}^1)}{\partial b_j}\Big)$ is given below.

$$\mathrm{Var}\Big(\frac{\partial \hat{f}(\mathbf{v}^1)}{\partial b_j}\Big) = \mathrm{Var}\Big(\frac{1}{N^1} \sum_{i=1}^{N^1} \frac{\partial \hat{f}(\mathbf{v}_i^1)}{\partial b_j}\Big)$$

(6)

Theorems 1-5 in Appendix A.5 provide the theoretical analysis about the density estimation, derivatives, and variances for well understanding the KDE technique.

In this work, assuming that the graph data follow the Gaussian distribution, a product Gaussian kernel $\mathcal{K}$ is used to estimate the node density $\hat{f}(\mathbf{v}^1)$. $\frac{\partial \hat{f}(\mathbf{v}^1)}{\partial b_j}$ is accordingly updated as follows.

$$\frac{\partial \hat{f}(\mathbf{v}^1)}{\partial b_j} = \frac{C}{N^1} \sum_{i=1}^{N^1} \Big((\mathbf{v}_j^1 - \mathbf{v}_{ij}^1)^2 - b_j^2\Big) \prod_{l=1}^{N^1} \mathcal{K}(\frac{\mathbf{v}_l^1 - \mathbf{v}_{il}^1}{b_l}) \propto \frac{1}{N_1} \sum_{i=1}^{N^1} \Big((\mathbf{v}_j^1 - \mathbf{v}_{ij}^1)^2 - b_j^2\Big) \prod_{l=1}^{N^1} \mathcal{K}(\frac{\mathbf{v}_l^1 - \mathbf{v}_{il}^1}{b_l})$$

$$= \frac{1}{N^1} \sum_{i=1}^{N^1} \Big((\mathbf{v}_j^1 - \mathbf{v}_{ij}^1)^2 - b_j^2\Big) \exp\big(-\sum_{l=1}^{N^1} \frac{(\mathbf{v}_l^1 - \mathbf{v}_{il}^1)^2}{2 b_j^2}\big)$$

(7)

where $C$ denotes a proportionality constant $C = \frac{1}{b_j^3} \prod_{l=1}^{N^1} \frac{1}{b_l}$. It can be safely ignored to avoid computation overflow when $b_l \to 0$ for large $N^1$. The bandwidth estimation is presented in Algorithm 1.

Based on the estimated $\mathbf{B}$ and the Gaussian kernel $\mathcal{K}$, the closed form of $\hat{f}(\mathbf{v}^1)$ is derived below.

$$\hat{f}(\mathbf{v}^1) = \frac{1}{N^1} \sum_{i=1}^{N^1} \prod_{j=1}^{N^1} \mathcal{K}(\frac{\mathbf{v}_j^1 - \mathbf{v}_{ij}^1}{b_j}) \sqrt{\frac{|\mathbf{B} + \boldsymbol{\Sigma}|}{|\boldsymbol{\Sigma}|}} \times \exp\big(-\frac{(\mathbf{v}^1 - \mu)^T (\boldsymbol{\Sigma}^{-1} - (\mathbf{B} + \boldsymbol{\Sigma})^{-1})(\mathbf{v}^1 - \mu)}{2}\big)$$

(8)

where $\mu$ and $\boldsymbol{\Sigma}$ are the maximum likelihood estimation of the mean vector and covariance matrix of the Gaussian distribution. Please refer to Appendices A.6 and A.7 for detailed derivation of $\hat{f}(\mathbf{v}^1)$.

As two graphs $G^1$ and $G^2$ often have different structures and distributions and thus the same KDE method as Algorithm 1 is utilized to estimate the density $\hat{g}(\mathbf{v}^2)$ of $\mathbf{v}^2$. Based on the estimations $\hat{f}(\mathbf{v}^1)$ and $\hat{g}(\mathbf{v}^2)$, the attacker aims to maximize the following loss $\mathcal{L}_D$ with imperceptible perturbations.

$$\mathcal{L}_D = \sum_{(\hat{\mathbf{v}}_i^1, \hat{\mathbf{v}}_k^2) \in D} \mathcal{L}(\hat{\mathbf{v}}_i^1, \hat{\mathbf{v}}_k^2) \text{ where } \mathcal{L}(\hat{\mathbf{v}}_i^1, \hat{\mathbf{v}}_k^2) = \|M(\hat{\mathbf{v}}_i^1) - \hat{\mathbf{v}}_k^2\|_2^2 + \hat{f}(\hat{\mathbf{v}}_i^1) + \hat{g}(\hat{\mathbf{v}}_k^2)$$

(9)

where $\hat{\mathbf{v}}_i^1 = \mathbf{v}_i^1 + \delta_i^1$ (and $\hat{\mathbf{v}}_k^2 = \mathbf{v}_k^2 + \delta_k^2$) denote adversarial versions of clean nodes $\mathbf{v}_i^1$ (and $\mathbf{v}_k^2$) in $G^1$ (and $G^2$) by adding a small amount of edge perturbations $\delta_i^1$ (and $\delta_k^2$) through our proposed MLPGD method in the next section, such that $M(\hat{\mathbf{v}}_i^1)$ is far away from $\hat{\mathbf{v}}_k^2$ and thus the matching accuracy is decreased. In addition, we push $\mathbf{v}_i^1$ and $\mathbf{v}_k^2$ to dense regions to generate $\hat{\mathbf{v}}_i^1$ and $\hat{\mathbf{v}}_k^2$, by maximizing $\hat{f}(\hat{\mathbf{v}}_i^1)$ and $\hat{g}(\hat{\mathbf{v}}_k^2)$, such that $\hat{\mathbf{v}}_i^1$ and $\hat{\mathbf{v}}_k^2$ are indistinguishable from their neighbors in perturbed graphs. This reduces the possibility of perturbation detection by humans or defender programs.

---

**Algorithm 1 Bandwidth Matrix Estimation**

**Input:** graph $G^1 = (V^1, E^1)$, parameter $0 < s < 1$, initial bandwidth $b_0$, and parameter $c$.
**Output:** Bandwidth matrix $\mathbf{B}$.
1: Initialize all $b_1, \cdots, b_{N^1}$ with $b_0$;
2: **for** each $j = 1$ **to** $N^1$
3:    **do**
4:       Estimate the derivative $\frac{\partial f(\mathbf{v}^1)}{\partial b_j}$ and variance $\mathrm{Var}(\frac{\partial f(\mathbf{v}^1)}{\partial b_j})$ in Eqs.(6)-(7);
5:       Compute the threshold $\tau_j = \sqrt{2 \cdot \mathrm{Var}(\frac{\partial f(\mathbf{v}^1)}{\partial b_j}) \cdot \log(cN^1)}$;
6:       **if** $\left|\frac{\partial f(\mathbf{v}^1)}{\partial b_j}\right| > \tau_j$, **then** Update $b_j = b_j s$;
7:    **while** $\left|\frac{\partial f(\mathbf{v}^1)}{\partial b_j}\right| > \tau_j$
8: **Return** $\mathbf{B}$.

---

**Algorithm 2 Meta Learning-based Projected Gradient Descent (MLPGD)**

**Input:** Batches $D_1, \cdots, D_C$ in a set $D$ of node pairs, initial general policy parameters $\{\theta^1, \theta^2\}$, adaptation step size $\alpha$, meta step size $\beta$.
**Output:** Optimized $\{\theta^1, \theta^2\}$.
1: **Repeat** until convergence
2:    Sample $C$ batches of anchor node pairs $D_1, \cdots, D_C$;
3:    **for** $c = 1$ **to** $C$
4:       Estimate gradient $\mathbf{e}_c = e(D_c, \{\theta^1, \theta^2\})$;
5:       Compute adapted parameters $\{\theta_c^1, \theta_c^2\} = \{\theta^1, \theta^2\} + \alpha \mathbf{e}_c$;
6:    Update parameters $\{\theta^1, \theta^2\} = \{\theta^1, \theta^2\} + \frac{\beta}{C} \sum_{c=1}^C e(D_c, \{\theta_c^1, \theta_c^2\})$;
7: **Return** $\{\theta^1, \theta^2\}$.

---

# 4 Effective Attacks via Meta Learning-based Projected Gradient Descent

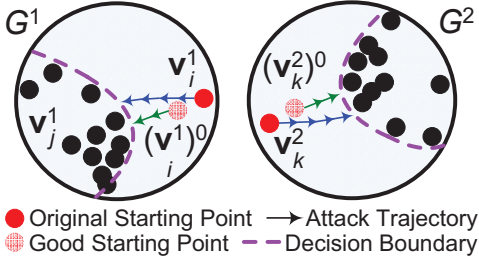

● Original Starting Point   → Attack Trajectory
⊕ Good Starting Point   – – Decision Boundary

Figure 2: Effective Attacks

In Figure 2, two dashed purple curves denote the decision boundary of graph matching. If we move a clean node $\mathbf{v}_i^1$ across the decision boundary to generate an adversarial node $\hat{\mathbf{v}}_i^1$, then we have other nodes $\mathbf{v}_j^1$ to make $M(\mathbf{v}_j^1)$ and $\mathbf{v}_k^2$ become more similar than $M(\hat{\mathbf{v}}_i^1)$ and $\mathbf{v}_k^2$, and thus a wrong matching $(\mathbf{v}_j^1, \mathbf{v}_k^2)$ will be produced. Blue and green polylines denote attack trajectories starting from original and good starting pints with gradient descent method respectively. A shortcut from good starting points $(\mathbf{v}_i^1)^0$ or $(\mathbf{v}_k^2)^0$ is able to cr-

oss the peak of the decision boundary and converge quickly, while the trajectories from the original nodes $\mathbf{v}_i^1$ or $\mathbf{v}_k^2$ take long walks to cross the non-peak boundary.

Based on the attack loss in Eq.(9), we propose to integrate meta learning and PGD into an MLPGD model, to produce more effective adversarial nodes with good starting points towards graph matching.

$$\begin{aligned}
(\mathbf{v}_i^1)^{(t+1)} &= \Pi_{\triangle_i^1} \mathrm{sgn}\left[\mathrm{ReLU}\left(\nabla_{(\mathbf{v}_i^1)^t} \mathcal{L}((\mathbf{v}_i^1)^t, (\mathbf{v}_k^2)^t)\right)\right] \\
(\mathbf{v}_k^2)^{(t+1)} &= \Pi_{\triangle_k^2} \mathrm{sgn}\left[\mathrm{ReLU}\left(\nabla_{(\mathbf{v}_k^2)^t} \mathcal{L}((\mathbf{v}_i^1)^t, (\mathbf{v}_k^2)^t)\right)\right], \quad t = 1, \cdots, T
\end{aligned} \tag{10}$$

where $(\mathbf{v}_i^1)^t$ and $(\mathbf{v}_k^2)^t$ denotes the adversarial nodes of $\mathbf{v}_i^1$ and $\mathbf{v}_k^2$ derived at step $t$. $\epsilon$ specifies the budget of allowed perturbed edges for each attacked node. $\triangle_i^1 = \{(\delta_i^1)^t | \mathbf{1}^T (\delta_i^1)^t \leq \epsilon, (\delta_i^1)^t \in \{0, 1\}^{N^1}\}$, where $(\delta_i^1)^t = \|(\mathbf{v}_i^1)^t - \mathbf{v}_i^1\|_2^2$, represents the constraint set of the projection operator $\Pi$, i.e., it encodes whether an edge of $\mathbf{v}_i^1$ is modified or not. $\triangle_k^2$ has the similar definition for $\mathbf{v}_k^2$. The composition of the ReLU and sign operators guarantees $(\mathbf{v}_i^1)^t \in \{0, 1\}^{N^1}$ and $(\mathbf{v}_k^2)^t \in \{0, 1\}^{N^2}$, as it adds (or removes) an edge or keeps it unchanged when an derivate in the gradient is positive (or negative). The outputs $(\mathbf{v}_i^1)^T$ and $(\mathbf{v}_k^2)^T$ at final step $T$ are used as the adversarial nodes $\hat{\mathbf{v}}_i^1$ and $\hat{\mathbf{v}}_k^2$.

Searching for attack starting points for each $(\mathbf{v}_i^1, \mathbf{v}_k^2)$ in large graphs is computationally inefficient. Meta learning techniques aim to train a general model with general parameters that can quickly adapt to a variety of new learning tasks with refined parameters [23, 3, 42, 56]. This offers a great opportunity to find good attack starting points $(\mathbf{v}_i^1)^0$ and $(\mathbf{v}_k^2)^0$ for all $(\mathbf{v}_i^1, \mathbf{v}_k^2) \in D$ with lower cost, such that the generated $\hat{\mathbf{v}}_i^1$ and $\hat{\mathbf{v}}_k^2$ by the PGD model can maximize the attack loss $\mathcal{L}_D$ in Eq.(9).

**Algorithm 3 Gradient Estimation** $e\big(D_c, \{\theta^1, \theta^2\}, N, \lambda\big)$

---

**Input:** Batch $D_c$, general parameters $\{\theta^1, \theta^2\}$, number of samples $N$ in Monte Carlo REINFORCE, smoothing parameter $\lambda$.
**Output:** Gradient estimation of $a$.
  1: Sample $N$ i.i.d. Gaussian matrices $\mathbf{g}_1, \cdots, \mathbf{g}_N \sim \mathcal{N}(0, \mathbf{I})$;
  2: **Return** gradient estimation $\frac{1}{N\lambda} \sum_{i=1}^{N} a\big(D_c, \{\theta^1, \theta^2\} + \lambda\mathbf{g}_i\big)\mathbf{g}_i$.

---

**Algorithm 4 Adversarial Attack** $a\big(D_c, \{\theta_c^1, \theta_c^2\}\big)$

---

**Input:** Batch $D_c$, perturbation budget $\epsilon$, specific parameters $\{\theta_c^1, \theta_c^2\}$
**Output:** Attack loss $\mathcal{L}_{D_c}$ on $D_c$.
  1: $\mathcal{L}_{D_c} = 0$;
  2: **for** each $(\mathbf{v}_i^1, \mathbf{v}_k^2) \in D_c$
  3:     Generate attack starting points $(\mathbf{v}_i^1)^0 = h^1\big(\mathbf{v}_i^1 | \theta_c^1\big)$ and $(\mathbf{v}_k^2)^0 = h^2\big(\mathbf{v}_k^2 | \theta_c^2\big)$;
  4:     Utilize PGD attack to generate adversarial nodes $(\mathbf{v}_i^1)^T$ and $(\mathbf{v}_k^2)^T$ in Eq.(10);
  5:     Aggregate attack loss $\mathcal{L}_{D_c} + = \mathcal{L}\big((\mathbf{v}_i^1)^T, (\mathbf{v}_k^2)^T\big)$ in Eq.(9);
  6: **Return** $\mathcal{L}_{D_c}$.

---

Algorithm 2 presents the pseudo code of our MLPGD model. $D$ is partitioned into $C$ batches $D_1, \cdots, D_C$, each with equal size of $|D|/C$. The search process on each batch $D_c$ $(1 \leq c \leq C)$ is treated as a single task, which aims to find good $(\mathbf{v}_i^1)^0$ and $(\mathbf{v}_k^2)^0$ for $D_c$ to maximize the attack loss $\mathcal{L}_{D_c} = \sum_{(\hat{\mathbf{v}}_i^1, \hat{\mathbf{v}}_k^2) \in D_c} \mathcal{L}(\hat{\mathbf{v}}_i^1, \hat{\mathbf{v}}_k^2)$. A general model that has general parameters $\theta^1, \theta^2$ is learnt to quickly adapt to search tasks on multiple batches. The learnt $\theta^1, \theta^2$ should be sensitive to changes of each $D_c$, such that small changes in $\theta^1, \theta^2$ will produce high rise on $\mathcal{L}_{D_c}$ over any of $D_1, \cdots, D_C$. Line 4 estimates the gradient of $\mathcal{L}_{D_c}$ by calling Algorithm 3. In Line 5, when adapting to the task on a new $D_c$, $\theta^1, \theta^2$ become specific parameters $\theta_c^1, \theta_c^2$ for $D_c$. Here, we use $\{\theta_c^1, \theta_c^2\}$ to denote the concatenation matrix of $\theta_c^1$ and $\theta_c^2$. The parameters are trained by maximizing the attack loss $a\big(D_c, \{\theta_c^1, \theta_c^2\}\big)$ w.r.t. general parameters $\theta^1, \theta^2$ across batches. The meta objective is given below.

$$\max \mathcal{L}_{D_c} = \max_{\theta^1, \theta^2} \sum_{c=1}^{C} a\big(D_c, \{\theta_c^1, \theta_c^2\}\big) = \sum_{c=1}^{C} a\big(D_c, \{\theta^1, \theta^2\} + \alpha\mathbf{e}_c\big) \tag{11}$$

In Line 6, the meta optimization is performed over the general $\theta^1, \theta^2$, while the objective is computed using the specific $\theta_c^1, \theta_c^2$. The general $\theta^1, \theta^2$ are updated in terms of the attack loss on each batch.

$$\{\theta^1, \theta^2\} = \{\theta^1, \theta^2\} + \frac{\beta}{C} \sum_{c=1}^{C} e\big(D_c, \{\theta_c^1, \theta_c^2\}\big) \tag{12}$$

Algorithm 4 exhibits the adversarial attack module $a\big(D_c, \{\theta_c^1, \theta_c^2\}\big)$ on a batch $D_c$ $(1 \leq c \leq C)$. In Line 3, two neural networks $h^1$ and $h^2$ with specific parameters $\theta_c^1$ and $\theta_c^2$ are designed to generate the attack starting points $(\mathbf{v}_i^1)^0$ and $(\mathbf{v}_k^2)^0$ of each $(\mathbf{v}_i^1, \mathbf{v}_k^2) \in D_c$. The last layers of $h^1$ and $h^2$ use the composition of the ReLU [53] and Softsign [25] as activation function to ensure $(\mathbf{v}_i^1)^0 \in \{0, 1\}^{N^1}$ and $(\mathbf{v}_k^2)^0 \in \{0, 1\}^{N^2}$. In Line 4, the PGD attack in Eq.(10) is utilized to generate the adversarial nodes $\hat{\mathbf{v}}_i^1$ and $\hat{\mathbf{v}}_k^2$. Line 5 calculates the attack loss $\mathcal{L}_{D_c}$ on $D_c$ to provide task-specific feedback.

Standard meta learning models utilizes gradient ascent/descent techniques to compute the updated parameters on new tasks [23, 3, 42, 56]. However, the attack module in Algorithm 4 is non-smooth and non-differential w.r.t. parameters $\theta^1, \theta^2, \theta_c^1$, and $\theta_c^2$, since the perturbation is a multi-step process as well as the projection at each step is non-differential. Therefore, Algorithm 3 is proposed to employ Gaussian smoothing technique to approximate a smoothed attack module.

$$\hat{a}\big(D_c, \{\theta^1, \theta^2\}\big) \approx (2\pi)^{-\frac{d}{2}} \int a\big(D_c, \{\theta^1, \theta^2\} + \lambda\mathbf{g}\big) \exp\big(-\frac{1}{2}\|\mathbf{g}\|_2^2\big) d\mathbf{g}$$
$$= \mathbb{E}_{\mathbf{g} \sim \mathcal{N}(0, \mathbf{I})} a\big(D_c, \{\theta^1, \theta^2\} + \lambda\mathbf{g}\big) \tag{13}$$

where $\hat{a}$ is the Gaussian smoothing of $a$ and differentiable everywhere. $\lambda$ is a smoothing parameter, and $d$ is the number of entries in $\{\theta^1, \theta^2\}$. $\mathbf{g} \sim \mathcal{N}(0, \mathbf{I})$ that has the same size as $\{\theta_c^1, \theta_c^2\}$ is interpreted as policy exploration directions, i.e., as perturbations in policy space to be explored. Thus, the policy perturbations in $\mathbf{g}$ are introduced to $\theta_c^1$ and $\theta_c^2$ respectively. $\hat{a}$ is obtained by perturbing $a$ at a given point along Gaussian directions and averaging the evaluations. And then, Algorithm 3 estimates the gradient of $\hat{a}$ via Monte Carlo REINFORCE method [79].

Table 2: Mismatching rate (%) with 5% perturbed edges

| Attack Model | AS | | | SNS | | | DBLP | | |
|---|---|---|---|---|---|---|---|---|---|
| | SNNA | CrossMNA | DGMC | SNNA | CrossMNA | DGMC | SNNA | CrossMNA | DGMC |
| Clean | 53.9 | 46.6 | 34.7 | 45.2 | 50.4 | 41.6 | 56.1 | 51.9 | 63.2 |
| Random | 57.5 | 49.9 | 37.6 | 48.8 | 52.0 | 46.8 | 59.8 | 54.0 | 68.8 |
| RL-S2V | 56.5 | 51.8 | 36.5 | 51.3 | 53.2 | 45.8 | 62.6 | 56.7 | 69.3 |
| Meta-Self | 63.1 | 55.1 | 45.0 | 55.1 | 64.8 | 51.3 | 65.7 | 63.7 | 73.3 |
| CW-PGD | 61.7 | 59.1 | 49.6 | 54.9 | 63.0 | 49.6 | 68.7 | 66.6 | 75.4 |
| GF-Attack | 57.9 | 53.7 | 39.5 | 52.9 | 59.6 | 47.9 | 64.9 | 61.1 | 69.1 |
| CD-ATTACK | 59.0 | 51.7 | 42.7 | 54.0 | 59.8 | 50.2 | 64.0 | 61.8 | 72.0 |
| GMA | **64.2** | **62.9** | **54.9** | **61.2** | **69.6** | **55.7** | **74.2** | **74.3** | **80.7** |

$$e\big(D_c, \{\theta^1, \theta^2\}\big) \approx \nabla_{\theta^1, \theta^2} \hat{a}\big(D_c, \{\theta^1, \theta^2\}\big) \approx (2\pi)^{-\frac{d}{2}} \int a\big(D_c, \{\theta^1, \theta^2\} + \lambda \mathbf{g}\big) \exp\big(-\frac{1}{2}\|\mathbf{g}\|_2^2\big) \mathbf{g} d\mathbf{g}$$

$$= \frac{1}{\lambda} \mathbb{E}_{\mathbf{g} \sim \mathcal{N}(0, \mathbf{I})} a\big(D_c, \{\theta^1, \theta^2\} + \lambda \mathbf{g}\big) \mathbf{g} \approx \frac{1}{N\lambda} \sum_{i=1}^{N} a\big(D_c, \{\theta^1, \theta^2\} + \lambda \mathbf{g}_i\big) \mathbf{g}_i, \ \mathbf{g}_i \sim \mathcal{N}(0, \mathbf{I}) \quad (14)$$

# 5 Experimental Evaluation

Table 1: Experiment Datasets

| Dataset | AS | | SNS | | DBLP | |
|---|---|---|---|---|---|---|
| Graph | v1 | v2 | Last.fm | LiveJournal | 2013 | 2014 |
| #Nodes | 10,900 | 11,113 | 5,682 | 17,828 | 28,478 | 26,455 |
| #Edges | 31,180 | 31,434 | 23,393 | 244,496 | 128,073 | 114,588 |
| #Matched Nodes | 6,462 | | 2,138 | | 4,000 | |

In this section, we will show the effectiveness of the GMA model in this work for deep graph matching tasks over three groups of datasets: social networks (SNS) [98], autonomous systems (AS) [2], and D-BLP coauthor graphs [1], as shown in Table 1.

**Baselines.** We compare the GMA model with six state-of-the-art graph attack models. Random Attack randomly adds and removes edges to generate perturbed graphs. RL-S2V [14, 123] generates adversarial attacks on graph data based on reinforcement learning. Meta-Self [125] is a poisoning attack model for node classification by using meta-gradients to solve the bilevel optimization problem. CW-PGD [86] developed a PGD topology attack to attack a predefined or a retrainable GNN. GF-Attack [5] attacks general learning methods by devising new loss and approximating the spectrum. CD-ATTACK [41] hides nodes in the community by attacking the graph autoencoder model. The majority of existing efforts focus on adversarial attacks on single graph learning. To our best knowledge, there are no other attack baselines on graph matching available. We replace the original losses in the baselines with the matching loss for fair comparison in the experiments.

**Variants of GMA model.** We evaluate four variants of GMA to show the strengths of different components. GMA-KDE only uses the KDE and density maximization to generate imperceptible attacks. GMA-PGD only utilizes the basic PGD [47] to produce effective attacks. GMA-MLPGD employs our proposed MLPGD model to well choose good attack starting points in the PGD. GMA operates with the full support of both KDE and MLPGD components.

**Graph matching algorithms.** We validate the effectiveness of the above attack models with three representative deep graph matching methods. SNNA [39] is an adversarial learning framework to solve the weakly-supervised identity matching problem by minimizing the distribution distance. CrossMNA [13] is a cross-network embedding-based supervised network alignment method by learning inter/intra-embedding vectors for each node and by computing pairwise node similarity scores across networks. Deep graph matching consensus (DGMC) [22] is a supervised graph matching method that reaches a data-driven neighborhood consensus between matched node pairs.

**Evaluation metrics.** We use two popular measures in graph matching to verify the attack quality: $Accuracy$ [93, 10, 95] and $Precision@K$ [101, 13, 97]. A larger mismatching rate (i.e., 1 - $Accuracy$ on test data) or a smaller $Precision@K$ shows a better attack. $K$ is fixed to 30 in all tests.

**Attack performance on various datasets with different matching algorithms.** Table 2 exhibits the mismatching rates of three deep graph matching algorithms on test data by eight attack models over three groups of datasets. We randomly sample 10% of known matched node pairs as training data and the rest as test data. For all attack models, the number of perturbed edges is fixed to 5% in these experiments. It is observed that among eight attack methods, no matter how strong the attacks are, the GMA method achieve the highest mismatching rates on perturbed graphs in most experiments, showing the effectiveness of GMA to the adversarial attacks. Compared to the graph matching results under other attack models, GMA, on average, achieves 21.3%, 18.8%, and 19.2%

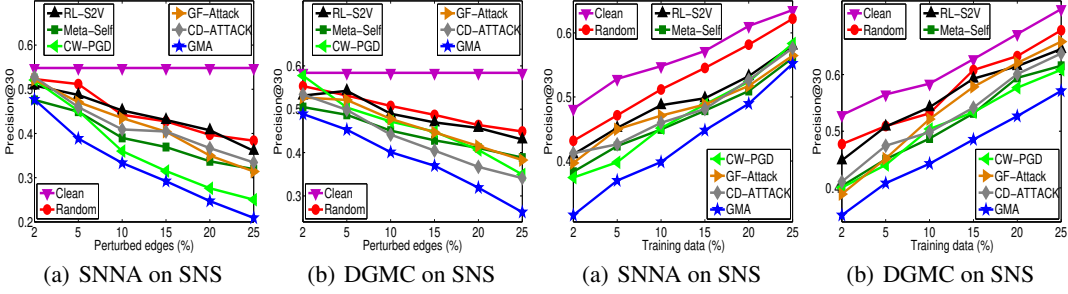

Figure 3: Precision with varying perturbed edges    Figure 4: Precision with varying training ratios

improvement of mismatching rates on AS, SNS, and DBLP respectively. In addition, the promising performance of GMA with all three graph matching models implies that GMA has great potential as a general attack solution to other graph matching methods, which is desirable in practice.

**Attack performance with varying perturbation edges.** Figure 3 presents the graph matching quality under eight attack models by varying the ratios of perturbed edges from 2% to 25%. It is obvious that the attacking performance improves for each attacker with an increase in the number of perturbed edges. This phenomenon indicates that current deep graph matching methods are very sensitive to adversarial attacks. GMA achieves the lowest $Precision$ values ($< 0.488$), which are still better than the other seven methods in most tests. Especially, when the perturbation ratio is large than 10%, the $Precision$ values drop quickly.

**Impact of training data ratios.** Figure 4 shows the quality of two graph matching algorithms on SNS by varying the ratio of training data from 2% to 25%. Here, the number of perturbed edges is fixed to 5%. We make the following observations on the performances by eight attack models. (1) The performance curves keep increasing when the training data ratio increases. (2) GMA outperforms other methods in most experiments with the lowest $Precision$: $< 0.482$ with SNNA and $< 0.571$ with DGMC respectively. Even when there are many training data available ($\geq 20\%$), the quality degradation by GMA is still obvious, although more training data makes the graph matching models be resilient to poisoning attacks under a small perturbation budget.

**Ablation study.** Figure 5 presents the mismatching rates of graph matching on SNS with four variants of the GMA attack model. We have observed the complete GMA achieves the highest mismatching rates ($> 54.9\%$) on AS, ($> 55.7\%$) over SNS, and ($> 74.2\%$) on DBLP, which are obviously better than other versions. Notice that GMA-MLPGD performs quite well in most experiments, compared with GMA-PGD. A reasonable explanation is that searching from good attack starting points can help the MLPGD converge quickly by crossing the peak of the decision boundary. In addition, GMA-KDE achieves the better attack performance than GMA-MLPGD. A rational guess is that it is difficult to correctly match two nodes results when they lie in dense regions with many similar nodes, although the main goal of KDE is to generate imperceptible attacks. These results illustrate both KDE and MLPGD models are important in producing effective attacks in graph matching.

**Impact of perturbation budget $\epsilon$.** Figure 6 (a) measures the performance effect of $\epsilon$ in the MLPGD model for the graph matching by varying $\epsilon$ from 1 to 5. It is observed that when increasing $\epsilon$, the $Precision$ of the GMA model decreases substantially. This demonstrates that it is difficult to train a robust graph matching model under large $\epsilon$ constraint. However, a large $\epsilon$ can be easily detected by humans or by defender programs. Notice that the average node degree of three groups of datasets is between 2.9 and 13.9. Thus we suggest generating both imperceptible and effective attacks for the graph matching task under $\epsilon$ between 2 and 3, such that $\epsilon$ is smaller than the average node degree.

**Time complexity analysis** Based on [20], the complexity of meta learning is $O(d^2)$, where $d$ is the problem dimension. In the context of graph matching, it is the number of nodes in two graphs ($N^s$, $s = 1$ or 2). Both density estimation and PGD have complexity of $O((N^s)^2)$. Thus, the overall complexity is $O((N^s)^2)$, which is the same as most existing attack methods that search the entire graphs to find weak edges to attack.

**Impact of meta step size $\alpha$.** Figure 6 (b) shows the impact of $\alpha$ in our MLPGD model over three groups of datasets. The performance curves initially raise when $\alpha$ increases. Intuitively, the MLPGD with large $\alpha$ can help the meta learning converge quickly. Later on, the performance curves keep relatively stable or even decreasing when $\alpha$ continuously increases. A reasonable explanation is that

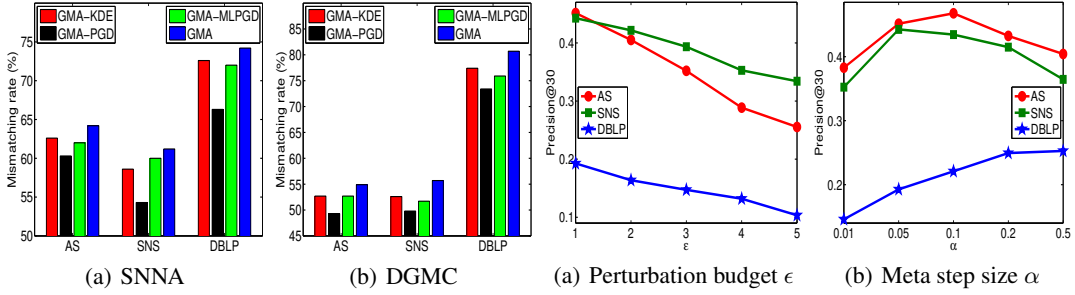

| (a) SNNA | (b) DGMC | (a) Perturbation budget $\epsilon$ | (b) Meta step size $\alpha$ |

Figure 5: Mismatching rate (%) of GMA variants    Figure 6: Precision with varying parameters

the too large $\alpha$ makes the meta learner take a big walk with rapid pace, such that it may miss the optimal meta parameters. Thus, it is important to determine the optimal $\alpha$ for the MLPGD model.

# 6    Related Work

**Adversarial Attacks on Graph Data.** Several recent studies have presented that graph learning models, especially deep learning-based models, are highly sensitive to adversarial attacks, i.e., carefully designed small deliberate perturbations in graph structure and attributes can cause the models to produce incorrect prediction results [64, 69, 90, 74, 45, 85, 80, 127]. The current graph adversarial attack techniques mainly fall into two categories in terms of the attack surface: (1) evasion attacks occur after the target model is well trained in clean graphs, i.e., the learned model parameters are fixed during evasion attacks. The attacker tries to evade the graph learning models by generating malicious samples during testing phase [14, 126]; and (2) poisoning attacks, known as contamination of the training data, take place during the training time of deep learning models. An adversary tries to poison the training data by injecting carefully designed examples to cause failures of the target model on some given test samples [126, 69, 125, 4, 123]. Since transductive learning is widely used in most graph analysis tasks, the test samples (but not their labels) are participated in the training stage, which leads to the popularity of poisoning attacks. Various adversarial attack models have been developed to show the vulnerability of graph learning models in node classification [14, 126, 74, 86, 125, 71, 19, 70], community detection [9, 78, 7, 41], network embedding [6, 4, 5], link prediction [104], similarity search [15], malware detection [30], and knowledge graph embedding [90].

**Graph Matching.** Graph data analysis has attracted active research in the last decade [110, 111, 11, 106, 12, 107, 108, 66, 113, 109, 112, 114, 35, 115, 117, 116, 119, 118, 81, 82, 120, 121]. Graph matching is one of the most important research topics in the graph domain, which aims to match the same entities (i.e., nodes) across two or more graphs and has been a heated topic in recent years [91, 98, 62, 46, 48, 101, 13]. Research activities can be classified into three broad categories. (1) Topological structure-based techniques, which rely on only the structural information of nodes to match multiple or two input networks, including IONE [43], GeoAlign [46], Low-rank EigenAlign [54], FRUI-P [105], CrossMNA [13], MOANA [96], GWL [84], MSUIL [38], DeepMGGE [24], and KEMINA [122]; (2) Structure and/or attribute-based approaches, which utilize highly discriminative structure and attribute features for ensuring the matching effectiveness, such as FINAL [93, 94], ULink [51], CAlign [10], MASTER [65], gsaNA [88], CoLink [100], REGAL [28], UUIL [37], SNNA [39], RANA [58], CENALP [18], ORIGIN [97], OPTANE [50], and Deep Graph Matching Consensus [22]; (3) Heterogeneous methods employ heterogeneous structural, content, spatial, and temporal features to further improve the matching performance, including COSNET [98], Factoid Embedding [83], HEP [99], LHNE [77], and DPLink [21]. Several papers review key achievements of graph matching across online information networks including state-of-the-art algorithms, evaluation metrics, representative datasets, and empirical analysis [62, 27, 31, 73].

# 7    Conclusions

In this work, we have studied the graph matching adversarial attack problem. First, we proposed to utilize kernel density estimation technique to estimate and maximize the densities of attacked nodes and generate imperceptible perturbations, by pushing attacked nodes to dense regions in two graphs. Second, we developed a meta learning based projected gradient descent method to well choose attack starting points and improve the search performance of PGD for producing effective perturbations. The GMA model achieves superior attack performance against several representative attack models.

## Broader Impact

Graph data are ubiquitous in the real world, ranging from biological, communication, and transportation graphs, to knowledge, social, and collaborative networks. Many real-world graphs are essentially crowdsourced projects, such as social and knowledge networks, where information and knowledge are produced by internet users who came to the sites. Thus, the quality of crowdsourced graph data is not stable, depending on human knowledge and expertise. In addition, it is well known that the openness of crowdsourced websites makes them vulnerable to malicious behaviors of interested parties to gain some level of control of the websites and steal users' sensitive information, or deliberately influence public opinion by injecting misleading information and knowledge into crowdsourced graphs.

Graph matching is one of the most important research topics in the graph domain, which aims to match the same entities (i.e., nodes) across two or more graphs [91, 98, 43, 46, 48, 72, 54, 105, 13, 75]. It has been widely applied to many real-world applications ranging from protein network matching in bioinformatics [33, 63], user account linking in different social networks [62, 51, 100, 37, 101, 21, 38], and knowledge translation in multilingual knowledge bases [87, 124], to geometric keypoint matching in computer vision [22]. Owing to the openness of crowdsourced graphs, more work is needed to analyze the vulnerability of graph matching under adversarial attacks and to future develop robust solutions that are readily applicable in production systems.

A potential downside of this research is about the application of user account linking in different social networks due to the user privacy issues. Recent advances in differential privacy and privacy preserving graph analytics have shown the superior performance of protecting sensitive information about individuals in the datasets. Therefore, these techniques offer a great opportunity to integrate them into the vulnerability analysis of graph matching, for alleviating the user privacy threats.

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
