[Supplementary Material]

# A Supplementary Materials

## A.1 Comparison with Existing Meta Learning-based Adversarial Attack Techniques

Meta-Self [125] is a poisoning attack model for node classification by leveraging meta-learning to generate attacks, i.e., using meta-gradients to solve the bilevel optimization problem. It conducts adversarial attacks on global node classification of a single graph. It aims to solve a bilevel optimization problem: (1) training classification on graphs and (2) attacking graphs. It gradually improves attack performance by using meta learning to iteratively solve the above two problems. The GMA model utilizes meta learning to find good attack starting points in two graphs.

## A.2 Supplementary Experiments on Transferability

Table 3: Experiment Datasets

| Dataset | Cora | Citeseer | Pubmed |
|---|---|---|---|
| #Nodes | 2,708 | 3,327 | 19,717 |
| #Edges | 5,429 | 4,732 | 44,338 |
| #Classes | 7 | 6 | 3 |

In this section, we use three popular real graph datasets in network embedding and node classification for supplementary experiments [S5, S6, S7, S8], as shown in Table 3.

**Transferability study.** In this paper, we also explore whether the GMA adversarial attack model can be applied to other graph learning tasks. In this vein, we select two representative graph applications (i.e., node classification and network embedding) with widely used target learning models: (1) graph convolutional network (GCN) generalizes convolutional neural networks on graph data to learn semi-supervised node classification [S2, S7]; and (2) GraphSAGE is a general framework for inductive node embedding [S1]. We use the link prediction algorithm based on common neighbors [S3] to predict the links between the nodes.

For the node classification task, the attacker aims to maximize the following loss $\mathcal{L}_D$ based on the cross-entropy loss.

$$\mathcal{L}_D = \sum_{\hat{\mathbf{v}}_i \in D} \Big( -\sum_{c=1}^{C} \mathbf{Y}_{\hat{\mathbf{v}}_i c} \log \tilde{\mathbf{Y}}_{\hat{\mathbf{v}}_i c} + \hat{f}(\hat{\mathbf{v}}_i) \Big) \tag{15}$$

where $C$ is the number of classes, $\mathbf{Y}$ is the label matrix and $\tilde{\mathbf{Y}} = softmax(\mathbf{H}^{(L)})$ are predictions of GCN by passing the hidden representation in the final layer $\mathbf{H}^{(L)}$ to a softmax function. $\hat{f}(\hat{\mathbf{v}}_i)$ is the kernel density estimation of node $\hat{\mathbf{v}}_i$.

The attack goal of the network embedding is to minimize the similarities between connected nodes while maximizing the similarities between isolated nodes by maximizing the following loss function.

$$\mathcal{L} = \sum_{\hat{\mathbf{v}}_i \in D} \Bigg( \sum_{\hat{\mathbf{v}}_j \in N(\hat{\mathbf{v}}_i)} \Big( -\log \sigma\big(E(\hat{\mathbf{v}}_i)^T \cdot E(\hat{\mathbf{v}}_j)\big) + \hat{f}(\hat{\mathbf{v}}_j)\Big) +$$
$$\sum_{k=1}^{K} \mathbb{E}_{\hat{\mathbf{v}}_k \sim p(\hat{\mathbf{v}}_k)} \Big( \log \sigma\big(E(\hat{\mathbf{v}}_i)^T \cdot E(\hat{\mathbf{v}}_k)\big) + \hat{f}(\hat{\mathbf{v}}_k)\Big) + \hat{f}(\hat{\mathbf{v}}_i) \Bigg) \tag{16}$$

where $E(\cdot)$ represents the GraphSAGE node embedding model. $E(\hat{\mathbf{v}}_i)^T$ is the transpose of $E(\hat{\mathbf{v}}_i)$. $p(\hat{\mathbf{v}}_k)$ denotes the distribution for sampling $K$ negative nodes $\hat{\mathbf{v}}_k \neq \hat{\mathbf{v}}_j$ through the negative sampling method [S4]. $\sigma(\cdot)$ is the sigmoid function. The inner product $\cdot$ represents the similarity degree between two embedding vectors. The above loss is equivalent to a cross-entropy loss with $(\hat{\mathbf{v}}_i, \hat{\mathbf{v}}_j)$ as positive samples and $(\hat{\mathbf{v}}_i, \hat{\mathbf{v}}_k)$ as negative ones.

Based on the above two loss functions, we utilize our proposed MLPGD model in Section 4 to generate adversarial nodes.

Table 4: Accuracy (%) with 5% perturbed edges

| Attack Model | Cora | | Citeseer | | Pubmed | |
|---|---|---|---|---|---|---|
| | GCN | GraphSage | GCN | GraphSage | GCN | GraphSage |
| Clean | 81.5 | 83.2 | 70.9 | 84.5 | 79.6 | 88.1 |
| Random | 78.3 | 81.6 | 69.6 | 83.5 | 78.7 | 85.2 |
| RL-S2V | 75.0 | 82.2 | 66.0 | 83.1 | 74.0 | 84.4 |
| Meta-Self | 77.3 | 78.1 | 68.8 | 82.3 | 77.6 | 82.3 |
| CW-PGD | **72.2** | 75.5 | 62.9 | 77.6 | 72.8 | 82.0 |
| GF-Attack | 72.6 | 74.1 | 64.7 | 77.1 | 73.0 | 79.5 |
| CD-ATTACK | 76.8 | 76.3 | 67.7 | 80.7 | 75.5 | 83.0 |
| GMA | 72.4 | **72.8** | **60.7** | **74.2** | **68.8** | **74.6** |

For the classification experiments, by following the setting in [126, 5] , we split the graph into labeled (20%) and unlabeled nodes (80%). Further, the labeled nodes are splitted into equal parts for training and validation. The attack performance is evaluated by the decrease of node classification accuracy. In the link prediction tests, we randomly hide 10% of the edges in the original graph as the positive samples. We also include equal number of randomly selected disconnected links that servers as negative samples. We use both positive and negative examples for evaluation. $Accuracy$ is used to quantify the quality of both node classification and link prediction. As we can see from Table 4, while all the attacking methods are effective, the GMA method achieves the best performance in most experiments. It validates the generalization ability of the GMA on other graph learning models.

## A.3 Supplementary Experiments on Graph Matching

**Validate attack performance under defenses.** We test two recent defense methods on generated attacks by the GMA model: [7] uses min-max adversarial training for defense and [8] vaccinates attack with low-rank approximations. As shown in Table 1, even with the defense, the GMA model can still achieve very high mismatching rate.

Table 5: Mismatching rate (%) with 5% perturbed edges

| Attack Model | AS | | | SNS | | | DBLP | | |
|---|---|---|---|---|---|---|---|---|---|
| | SNNA | CrossMNA | DGMC | SNNA | CrossMNA | DGMC | SNNA | CrossMNA | DGMC |
| Clean | 53.9 | 46.6 | 34.7 | 45.2 | 50.4 | 41.6 | 56.1 | 51.9 | 63.2 |
| GMA+Robust Training [86] | 62.6 | 58.5 | 53.0 | 56.2 | 66.5 | 51.8 | 71.8 | 71.0 | 77.3 |
| Vaccinated GMA [19] | 62.1 | 59.9 | 53.2 | 58.7 | 67.6 | 54.6 | 70.4 | 68.6 | 74.9 |
| GMA | **64.2** | **62.9** | **54.9** | **61.2** | **69.6** | **55.7** | **74.2** | **74.3** | **80.7** |

In the following experiments, unless otherwise explicitly stated, we use the DMGC model as the graph matching method in this section.

(a) Meta step size $\beta$    (b) #Samples $N$    (c) Initial bandwidth $b_0$    (d) Parameter $s$

Figure 7: Precision with varying parameters

**Impact of meta step size $\beta$.** Figure 7 (a) shows the impact of $\beta$ in our MLPGD model over three groups of datasets by varying $b_0$ from 0.01 to 0.5. We have witnessed the similar trend to Figure 6 (b). The performance curves initially raise and then drop quickly when $\beta$ continuously increases. This demonstrates that there must exist the optimal $\beta$ that makes the meta learning be maximally

optimized. We have observed that the *Precision* scores oscillate within the range of 47.0%, 33.8%, and 53.0% on AS, SNS, DBLP respectively.

**Sensitivity of number of samples $N$.** Figure 7 (b) exhibits the sensitivity of $N$ in our MLPGD model with $N$ between 1 and 15. In this paper, we estimate the gradient of the adversarial attack module $a$ in Algorithm 4 by perturbing $a$ at a given point along Gaussian directions and averaging the evaluations. The performance curves continuously increase with increasing $N$. This is consistent with the fact that increasing the sample size can reduce the bias in the gradient estimation as well as increase the diversity, so as to increase the probability of the estimator being close to the population gradient.

**Impact of initial bandwidth $b_0$.** Figure 7 (c) shows the impact of $b_0$ in the KDE model by varying $b_0$ from 0.5 to 2.5. the *Precision* values have concave curves when increasing $b_0$. This demonstrates that the initial bandwidth of the kernel has a strong influence on the resulting estimate. A reasonable observation is that the density estimation is undersmoothed since it results in too many spurious data artifacts arising by using a small $b_0$. On the other hand, the density estimation is oversmoothed since a large $b_0$ obscures much of the underlying structure. Therefore, there should exist the optimal $b_0$ that makes the estimation be optimally smoothed.

**Influence of parameter $s$.** Figure 7 (d) presents the influence of $s$ in the KDE model with $s$ between 0.1 and 0.9. It is observed that the performance first keeps stable or slightly increasing and then drops sharply with increasing $s$. In the KDE model, the bandwidth matrix $\mathbf{B}$ has strong influence on the density estimation. A good $\mathbf{B}$ should be as small as the data can allow. Thus, a smaller $s$ can help achieve a smaller $\mathbf{B}$ and make the KDE converge quickly. However, a too small $s$ may laead to the undersmoothed density estimation.

(a) SNNA        (b) DMGC

Figure 8: Mismatching rate (in %) with increasing iterations

**Convergence study.** Figure 8 presents the convergence of the adversarial attack training with SNNA and DMGC as the graph matching methods on SNS. As we can see, the mismatching rates keep increasing when we iteratively perform the attack task. The method converges when the numbers of iterations go beyond some thresholds. We have observed that the curves on AS and SNS converge gracefully within 500 iterations. However, the training over DBLP needs more iterations to achieve the convergence. This verifies the effectiveness of the GMA method based on the KDE and MLPGD techniques.

**Gaussian model on parameter estimation.** The Gaussian distribution is common in nature. Many papers assume graph representations follow Gaussian distributions, which allows to capture graph dynamics and uncertainty [S8-S11]. We have run the Shapiro-Wilk test [1], where if the P-Value of test $> 0.05$, the data is thought to follow a Gaussian distribution. In the test, the P-Values on AS and SNS are 0.668 and 0.543, which indicate two graphs follow Gaussian distribution.

## A.4 Experimental Details

**Environment.** The experiments were conducted on a compute server running on Red Hat Enterprise Linux 7.2 with 2 CPUs of Intel Xeon E5-2650 v4 (at 2.66 GHz) and 8 GPUs of NVIDIA GeForce GTX 2080 Ti (with 11GB of GDDR6 on a 352-bit memory bus and memory bandwidth in the neighborhood of 620GB/s), 256GB of RAM, and 1TB of HDD. Overall, the experiments took about 5 days in a shared resource setting. We expect that a consumer-grade single-GPU machine (e.g., with

a 1080 Ti GPU) could complete the full set of experiments in around 10 days, if its full resources were dedicated. The codes were implemented in Python 3.7.3 and PyTorch 1.0.14. We also employ Numpy 1.16.4 and Scipy 1.3.0 in the implementation.

**Implementation.** For three deep graph matching models of SNNA [2], CrossMNA [3], and DGMC [4], we used the open-source implementation and default parameter settings by the original authors for the experiments. All models were trained for 500 iterations, with a batch size of 512, and a learning rate of 0.001. For two other deep graph learning models of GCN [5] and GraphSAGE [6], we also use the default parameters in the authors' implementation. We used the public TensorFlow implementation of GCN and pass the hidden representation in the final layer by the GCN to a softmax function as the node classification results. We used the open-source TensorFlow implementation of GraphSAGE. To run GraphSAGE, it needs to train on an example graph or set of graphs. After training, GraphSAGE can be used to generate node embeddings for previously unseen nodes or entirely new input graphs, as long as these graphs have the same attribute schema as the training data. In the experiments, we use the node degree as node attributes. For five state-of-the-art graph attack models of RL-S2V [7], Meta-Self [8], CW-PGD [9], GF-Attack [10], and CD-ATTACK [11], we also utilized the same model architecture as the official implementation provided by the original authors and used the same perturbation budgets to attack the deep graph learning models in all experiments.

For the GMA attack model, we performed hyperparameter selection by performing a parameter sweep on initial adaptation step sizes $\alpha \in \{0.01, 0.05, 0.1, 0.2, 0.5\}$, meta step sizes $\beta \in \{0.01, 0.05, 0.1, 0.2, 0.5\}$, samples numbers $N \in \{1, 2, 5, 10, 15\}$, initial bandwidth $b_0 \in \{0.5, 1.0, 1.5, 2.0, 2.5\}$, and parameter $s \in \{0.1, 0.3, 0.5, 0.7, 0.9\}$. We select the best parameters over 50 iterations of training and evaluate the model at test time. In addition, two neural networks $h^1$ and $h^2$ in the adversarial attack module in Algorithm 4 are used to generate the good attack starting points. Both $h^1$ and $h^2$ are always implemented as three-layer perceptrons (input-hidden-output). The number of neurons in the hidden layer is set to 512. The model uses a mini-batch of size 512. The learning rate is equal to 0.001.

Notice that $\epsilon$ specifies the budget of allowed perturbed edges for each attacked node. Thus, $\epsilon$ should be a positive integer. In addition, most of real-world graph datasets, e.g., all datasets used in this paper are extremely sparse, i.e., the average node degree of most datasets in this paper is $2 \sim 5$. Thus, even if $\epsilon$ is very small, say 1 or 2, a large number of edges will be modified, which results in noticeable perturbations. For example, AS v1 contains 10,900 nodes and 31,180 edges. Therefore, we combine $\epsilon$ and the number limit of perturbed edges in entire graphs for actual perturbation budget. We set $\epsilon = 1$ and randomly select one target node to add or remove one edge at a time and repeat the same process to attack other nodes until the overall edge perturbations in entire graphs are beyond the number limit of perturbed edges, say 5%. After that, we stop to attack the rest of nodes.

## A.5 Theoretical Proof

**Assumption 1** *Assuming that $\mathcal{K}$ is a product symmetric kernel that satisfies $\int \mathcal{K}(\mathbf{u})d\mathbf{u} = 1$ and $\int \mathbf{u}\mathcal{K}(\mathbf{u})d\mathbf{u} = \mathbf{0}_{N^1}$, then*

$$\int \mathbf{u}\mathbf{u}^T \mathcal{K}(\mathbf{u})d\mathbf{u} = a\mathbf{I}_{N^1} \text{ and } a < \infty \tag{17}$$

$$\int \mathcal{K}^2(\mathbf{u})d\mathbf{u} = R(\mathcal{K}) < \infty \tag{18}$$

**Theorem 1** *Let $\mathbf{H}$ be the Hessian matrix of $f(\mathbf{v}^1)$, if the kernel density estimation $\hat{f}(x)$ of $f(x)$ is defined as follows.*

---

[2]https://github.com/yiweizhang526/social-network-alignment
[3]https://github.com/ChuXiaokai/CrossMNA
[4]https://github.com/rusty1s/deep-graph-matching-consensus
[5]https://github.com/tkipf/gcn
[6]http://snap.stanford.edu/graphsage/
[7]https://github.com/Hanjun-Dai/graph_adversarial_attack
[8]https://github.com/danielzuegner/gnn-meta-attack
[9]https://github.com/KaidiXu/GCN_ADV_Train
[10]https://github.com/SwiftieH/GFAttack
[11]https://github.com/halimiqi/CD-ATTACK

$$\hat{f}(\mathbf{v}^1) = \frac{1}{N^1 \det(\mathbf{B})} \sum_{i=1}^{N^1} \mathcal{K}(\mathbf{B}^{-1}(\mathbf{v}^1 - \mathbf{v}_i^1)) = \frac{1}{N^1} \sum_{i=1}^{N^1} \prod_{j=1}^{N^1} \frac{1}{b_j} \mathcal{K}\Big(\frac{\mathbf{v}_j^1 - \mathbf{v}_{ij}^1}{b_j}\Big), \tag{19}$$

*then*

$$\mathbf{E}(\hat{f}(\mathbf{v}^1)) = f(\mathbf{v}^1) + \frac{1}{2}a\big(tr(\mathbf{B}^T\mathbf{H}\mathbf{B})\big) + o_P\big(tr(\mathbf{B}^T\mathbf{B})\big) \tag{20}$$

*and*

$$\mathbf{Var}(\hat{f}(\mathbf{v}^1)) = \frac{1}{N^1 \det(\mathbf{B})} R(\mathcal{K}) f(\mathbf{v}^1) + o_P\Big(\frac{1}{N^1 \det(\mathbf{B})}\Big) \tag{21}$$

*Proof.*

$$
\begin{aligned}
\mathbf{E}(\hat{f}(\mathbf{v}^1)) &= \mathbf{E}\Big(\frac{1}{N^1 \det(\mathbf{B})} \sum_{i=1}^{N^1} \mathcal{K}(\mathbf{B}^{-1}(\mathbf{v}^1 - \mathbf{v}_i^1))\Big) \\
&= \frac{1}{\det(\mathbf{B})} \int \mathcal{K}(\mathbf{B}^{-1}(\mathbf{u} - \mathbf{v}^1)) f(\mathbf{u}) d\mathbf{u} \\
&= \int \mathcal{K}(\mathbf{u}) f(\mathbf{v}^1 + \mathbf{B}\mathbf{u}) d\mathbf{u} \\
&= \int \mathcal{K}(\mathbf{u})\Big(f(\mathbf{v}^1) + \mathbf{u}^T\mathbf{B}^T\nabla f(\mathbf{v}^1) + \frac{1}{2}\mathbf{u}^T\mathbf{B}^T\mathbf{H}\mathbf{B}\mathbf{u} + o_P\big(tr(\mathbf{u}^T\mathbf{B}^T\mathbf{B}\mathbf{u})\big)\Big) d\mathbf{u} \\
&= f(\mathbf{v}^1) + \frac{1}{2}a\big(tr(\mathbf{B}^T\mathbf{H}\mathbf{B})\big) + o_P\big(tr(\mathbf{B}^T\mathbf{B})\big)
\end{aligned}
\tag{22}
$$

*where $tr(\mathbf{X})$ denotes the trace of a matrix $\mathbf{X}$. $\mathbf{Y} = o_P(\mathbf{X})$ means $\mathbf{Y}/\|\mathbf{X}\|$ converges in probability to zero.*

*Since all $\mathbf{v}_i^1$s are independent and identically distributed, we have*

$$
\begin{aligned}
\mathbf{Var}(\hat{f}(\mathbf{v}^1)) &= \mathbf{Var}\Big(\frac{1}{N^1 \det(\mathbf{B})} \sum_{i=1}^{N^1} \mathcal{K}(\mathbf{B}^{-1}(\mathbf{v}^1 - \mathbf{v}_i^1))\Big) \\
&= \frac{1}{N^1 \det(\mathbf{B})^2} \mathbf{Var}\Big(\mathcal{K}(\mathbf{B}^{-1}(\mathbf{v}^1 - \mathbf{v}_i^1))\Big) \\
&= \frac{1}{N^1 \det(\mathbf{B})^2} \mathbf{E}\Big(\mathcal{K}^2(\mathbf{B}^{-1}(\mathbf{v}^1 - \mathbf{v}_i^1))\Big) - \frac{1}{N^1 \det(\mathbf{B})^2} \mathbf{E}^2\Big(\mathcal{K}(\mathbf{B}^{-1}(\mathbf{v}^1 - \mathbf{v}_i^1))\Big) \\
&= \frac{1}{N^1 \det(\mathbf{B})^2} \int \Big(\mathcal{K}(\mathbf{B}^{-1}(\mathbf{u} - \mathbf{v}^1))\Big)^2 f(\mathbf{u}) d\mathbf{u} - \frac{1}{N^1} \mathbf{E}^2(\hat{f}(\mathbf{v}^1)) \\
&= \frac{1}{N^1 \det(\mathbf{B})} \int (\mathcal{K}(\mathbf{u}))^2 f(\mathbf{v}^1 + \mathbf{B}\mathbf{u}) d\mathbf{u} - \frac{1}{N^1} \mathbf{E}^2(\hat{f}(\mathbf{v}^1)) \\
&= \frac{1}{N^1 \det(\mathbf{B})} R(\mathcal{K}) f(\mathbf{v}^1) + o_P\Big(\frac{1}{N^1 \det(\mathbf{B})}\Big)
\end{aligned}
\tag{23}
$$

*The Taylor series is utilized for expansion to generate the last equality.*

**Theorem 2** *The kernel density estimation $\hat{f}(x)$ converges towards the Gaussian distribution $\mathcal{N}(0, R(\mathcal{K})f(\mathbf{v}^1))$, i.e.,*

$$\sqrt{N^1\det(\mathbf{B})}\Big(\hat{f}(\mathbf{v}^1) - \mathbf{E}(\hat{f}(\mathbf{v}^1))\Big) \xrightarrow{d} \mathcal{N}(0, R(\mathcal{K})f(\mathbf{v}^1)) \tag{24}$$

*Proof. For ease of representation, a random variable $y_i$ is used to replace $\frac{\mathcal{K}\big(\mathbf{B}^{-1}(\mathbf{v}^1 - \mathbf{v}_i^1)\big)}{\det(\mathbf{B})}$. The skewness $s_{i3}$ of $y_i$ is calculated as follows.*

$$
\begin{aligned}
s_{i3} &= \mathbf{E}\big(|y_i - \mathbf{E}(y_i)|^3\big) \leq 8\mathbf{E}\big(|y_i|^3\big) = 8\mathbf{E}\Big(\Big|\frac{\mathcal{K}(\mathbf{B}^{-1}(\mathbf{v}^1 - \mathbf{v}_i^1))}{\det(\mathbf{B})}\Big|^3\Big) \\
&= \frac{8}{\det(\mathbf{B})^2} \int |\mathcal{K}(\mathbf{u})|^3 f(\mathbf{v}^1 + \mathbf{B}\mathbf{u}) d\mathbf{u} + +o_P\Big(\frac{1}{\det(\mathbf{B})^2}\Big)
\end{aligned}
\tag{25}
$$

*By using the same strategy in Eq.(23), we have*

$$\mathbf{Var}(y_i) = \frac{R(\mathcal{K})f(\mathbf{v}^1)}{\det(\mathbf{B})} + o_P\Big(\frac{1}{\det(\mathbf{B})}\Big) \tag{26}$$

*Based on the skewness $s_{i3}$, the Lyapunov condition at the $\rho^{th}$ moment with $\rho \geq 3$ is generated below.*

$$
\begin{aligned}
\frac{\big(\sum_{i=1}^{N^1} s_{i3}\big)^{\frac{1}{\rho}}}{\big(\sum_{i=1}^{N^1} \mathbf{Var}(y_i)\big)^{\frac{1}{\rho}}} &\leq \frac{\Big(\frac{8N^1}{\det(\mathbf{B})^2}\int |\mathcal{K}(\mathbf{u})|^3 f(\mathbf{v}^1 + \mathbf{B}\mathbf{u})d\mathbf{u} + + o_P\big(\frac{N^1}{\det(\mathbf{B})^2}\big)\Big)^{\frac{1}{3}}}{\Big(\frac{N^1 R(\mathcal{K})f(\mathbf{v}^1)}{\det(\mathbf{B})} + o_P\big(\frac{N^1}{\det(\mathbf{B})}\big)\Big)^{\frac{1}{2}}} \\
&= \frac{O\Big(\frac{(N^1)^{\frac{1}{3}}}{\det(\mathbf{B})^{\frac{2}{3}}}\Big)}{O\Big(\frac{(N^1)^{\frac{1}{2}}}{\det(\mathbf{B})^{\frac{1}{2}}}\Big)} = O\Big((N^1\det(\mathbf{B}))^{-\frac{1}{6}}\Big)
\end{aligned} \tag{27}
$$

*Thus, we get*

$$\lim_{N^1 \to \infty} \frac{\big(\sum_{i=1}^{N^1} s_{i3}\big)^{\frac{1}{\delta}}}{\big(\sum_{i=1}^{N^1} \mathbf{Var}(y_i)\big)^{\frac{1}{\delta}}} = 0 \tag{28}$$

*In terms of the Lyapunov central limit theorem, we have*

$$\frac{\hat{f}(\mathbf{v}^1) - \mathbf{E}\big(\hat{f}(\mathbf{v}^1)\big)}{\sqrt{\mathbf{Var}(\hat{f}(\mathbf{v}^1))}} \xrightarrow{d} \mathcal{N}(0,1) \tag{29}$$

*Based on the variance in Eq.(23), the proof is concluded.*

**Theorem 3** *If $\mathcal{K}$ is a product symmetric kernel with bandwidth matrix $\mathbf{B} = diag(b_1, \cdots, b_{N^1})$, then*

$$\mu_j = \frac{\partial}{\partial b_j}\mathbf{E}\big(\hat{f}(\mathbf{v}^1) - f(\mathbf{v}^1)\big) = ab_j\mathbf{H}_{jj} + o_P(b_j) \tag{30}$$

*Proof.*

*Based on Theorem 1, we have*

$$\mathbf{E}\big(\hat{f}(\mathbf{v}^1) - f(\mathbf{v}^1)\big) = \frac{1}{2}a\big(tr(\mathbf{B}^T\mathbf{H}\mathbf{B})\big) + o_P\big(tr(\mathbf{B}^T\mathbf{B})\big) \tag{31}$$

*Thus, the derivative can be directly generated.*

$$\mu_j = \frac{\partial}{\partial b_j}\mathbf{E}\big(\hat{f}(\mathbf{v}^1) - f(\mathbf{v}^1)\big) = ab_j\mathbf{H}_{jj} + o_P(b_j) \tag{32}$$

**Theorem 4** *If $\mathcal{K}$ is a product symmetric kernel with bandwidth matrix $\mathbf{B} = diag(b_1, \cdots, b_{N^1})$, then*

$$(\sigma_j)^2 = \mathbf{Var}\big(\frac{\partial \hat{f}(\mathbf{v}^1)}{\partial b_j}\big) = \frac{R(\mathcal{K})f(\mathbf{v}^1)}{4N^1(b_j)^2}\Big(\prod_{l=1}^{N^1}\frac{1}{b_l}\Big)(1 + o_P(1)) \tag{33}$$

*Proof. Based on Theorems 1 and 2, suppose that $\varphi \sim \mathcal{N}(0,1)$, the kernel density estimator $\hat{f}(\mathbf{v}^1)$ is denoted as follows.*

$$
\begin{aligned}
\hat{f}(\mathbf{v}^1) &= \mathbf{E}\big(\hat{f}(\mathbf{v}^1)\big) + \varphi\sqrt{\mathbf{Var}(\hat{f}(\mathbf{v}^1))} \\
&= f(\mathbf{v}^1) + \frac{1}{2}a\big(tr(\mathbf{B}^T\mathbf{H}\mathbf{B})\big) + o_P\big(tr(\mathbf{B}^T\mathbf{B})\big) + \varphi\sqrt{\mathbf{Var}(\hat{f}(\mathbf{v}^1))}
\end{aligned} \tag{34}
$$

*Therefore, we have*

$$\frac{\partial \hat{f}(\mathbf{v}^1)}{\partial b_j} = \frac{\partial f(\mathbf{v}^1)}{\partial b_j} + \frac{\partial}{\partial b_j}\Big(\frac{1}{2}a\big(tr(\mathbf{B}^T\mathbf{H}\mathbf{B})\big)\Big) + \frac{\partial}{\partial b_j}\Big(\varphi\sqrt{\mathbf{Var}(\hat{f}(\mathbf{v}^1))}\Big) \tag{35}$$

*As*

$$
\begin{aligned}
\frac{\partial}{\partial b_j}\Big(\sqrt{\mathbf{Var}(\hat{f}(\mathbf{v}^1))}\Big) &= \frac{1}{2\sqrt{\mathbf{Var}(\hat{f}(\mathbf{v}^1))}}\frac{\partial}{\partial b_j}\Big(\mathbf{Var}(\hat{f}(\mathbf{v}^1))\Big) \\
&= -\frac{1}{2\sqrt{\mathbf{Var}(\hat{f}(\mathbf{v}^1))}}\frac{R(\mathcal{K})f(\mathbf{v}^1)}{N^1 b_j \det(\mathbf{B})}\Big(1 + o_P\Big(\frac{1}{N^1 b_j \det(\mathbf{B})}\Big)\Big) \\
&= -\frac{1}{2}\sqrt{\frac{R(\mathcal{K})f(\mathbf{v}^1)}{N^1(b_j)^2\det(\mathbf{B})}}\big(1 + o_P(\sqrt{b_l})\big)
\end{aligned} \tag{36}
$$

*Thus,*

$$(\sigma_j)^2 = \mathbf{Var}\Big(\frac{\partial \hat{f}(\mathbf{v}^1)}{\partial b_j}\Big) = \frac{R(\mathcal{K})f(\mathbf{v}^1)}{4N^1(b_j)^2}\Big(\prod_{l=1}^{N^1}\frac{1}{b_l}\Big)(1 + o_P(1)) \tag{37}$$

**Theorem 5** *Recall the definition in Eq.(5), $\frac{\partial \hat{f}(\mathbf{v}^1)}{\partial b_j} = \frac{1}{N^1}\sum_{i=1}^{N^1}\frac{\partial \hat{f}(\mathbf{v}_i^1)}{\partial b_j}$, let $(\upsilon_j)^2$ be the sample variance of $\frac{\partial \hat{f}(\mathbf{v}_i^1)}{\partial b_j}(i = 1, \cdots, N^1)$, then $\frac{\partial \hat{f}(\mathbf{v}^1)}{\partial b_j}/\upsilon_j \sim \mathcal{N}(\omega, 1)$ where $\omega = \mu_j/\sigma_j$.*

*Proof. Based on the Khinchin's law, which states that the sample average converges in probability towards the expected value when $N^1 \to \infty$, we have*

$$\frac{\partial \hat{f}(\mathbf{v}^1)}{\partial b_j} = \frac{1}{N^1}\sum_{i=1}^{N^1}\frac{\partial \hat{f}(\mathbf{v}_i^1)}{\partial b_j} \xrightarrow{P} \mu_j \tag{38}$$

*Due to the consistency of the sample variance, we get*

$$(\upsilon_j)^2 \longrightarrow (\sigma_j)^2 \tag{39}$$

*$\frac{\partial \hat{f}(\mathbf{v}^1)}{\partial b_j}$ and $\upsilon_j$ are sequences of random variables. According to the Slutsky's theorem, if $\frac{\partial \hat{f}(\mathbf{v}^1)}{\partial b_j}$ converges in distribution to $\mu_j$ and $\upsilon_j$ converges in probability to $\sigma_j$, then*

$$\frac{\partial \hat{f}(\mathbf{v}^1)}{\partial b_j}/\upsilon_j \xrightarrow{d} \mu_j/\sigma_j \tag{40}$$

*Therefore, the proof is concluded.*

### A.6 The Closed Form of the Density Estimation $f(\mathbf{v}^1)$ with the Product Gaussian Kernel

In this work, we assume that the graph data follow the Gaussian distribution, a product Gaussian kernel $\mathcal{K}$ is used to estimate the node density $\hat{f}(\mathbf{v}^1)$. For ease of representation, let $K(\cdot) = \frac{1}{\det(\mathbf{B})}\mathcal{K}\left(\mathbf{B}^{-1}\cdot\right)$. We propose a parametric density estimation method to estimate the closed form of $\hat{f}(\mathbf{v}^1)$.

$$\hat{f}(\mathbf{v}^1) = \frac{\sum_{i=1}^{N^1} K(\mathbf{v}^1 - \mathbf{v}_i^1)\tilde{f}(\mathbf{v}^1)}{N^1 \int K(\mathbf{v}^1 - \mathbf{u})\tilde{f}(\mathbf{u})d\mathbf{u}}, \tag{41}$$

where $\tilde{f}(\mathbf{x}) = \mathcal{N}(\mu, \boldsymbol{\Sigma})$, $K(\mathbf{x}) = \mathcal{N}(0, \mathbf{B})$, $\mathbf{B} = diag(b_1^2, \ldots, b_{N^1}^2)$, $\boldsymbol{\Sigma} = diag(\sigma_1^2, \ldots, \sigma_{N^1}^2)$

where $\tilde{f}(\mathbf{v}^1)$ is the multivariate Gaussian density function with the parameters of $\mu$ and $\boldsymbol{\Sigma}$ as its parametric density estimator at point $\mathbf{v}^1$. $\mu$ and $\boldsymbol{\Sigma}$ are the maximum likelihood estimation of the mean vector and covariance matrix of the Gaussian distribution. $K(\mathbf{v}^1)$ is another multivariate Gaussian density function with the parameters of $0$ as the mean and the estimated $\mathbf{B}$ as the covariance in Algorithm 1. Namely, we get

$$\tilde{f}(\mathbf{x}) = \frac{1}{\sqrt{2\pi|\boldsymbol{\Sigma}|}} \exp\Big(-\frac{(\mathbf{x}-\mu)^T\boldsymbol{\Sigma}^{-1}(\mathbf{x}-\mu)}{2}\Big)$$
$$K(\mathbf{x}) = \frac{1}{\sqrt{2\pi|\mathbf{B}|}} \exp\Big(-\frac{\mathbf{x}^T\mathbf{B}^{-1}\mathbf{x}}{2}\Big) \tag{42}$$

Based on the convolution theorem, we know that the convolution of two Gaussian distribution functions is still a Gaussian probability density function with mean $\mu + 0$ and variance $\boldsymbol{\Sigma} + \mathbf{B}$. Thus, we have

$$N^1 \int K(\mathbf{v}^1 - \mathbf{u})\tilde{f}(\mathbf{u})d\mathbf{u} = \frac{N^1}{\sqrt{2\pi|\mathbf{B}+\boldsymbol{\Sigma}|}} \exp\Big(-\frac{(\mathbf{v}^1-\mu)^T(\mathbf{B}+\boldsymbol{\Sigma})^{-1}(\mathbf{v}^1-\mu)}{2}\Big) \tag{43}$$

Thus, we get

$$\frac{\tilde{f}(\mathbf{v}^1)}{N^1 \int K(\mathbf{v}^1 - \mathbf{u})\tilde{f}(\mathbf{u})d\mathbf{u}} = \frac{\frac{1}{\sqrt{2\pi|\boldsymbol{\Sigma}|}}\exp\Big(-\frac{(\mathbf{v}^1-\mu)^T\boldsymbol{\Sigma}^{-1}(\mathbf{v}^1-\mu)}{2}\Big)}{\frac{N^1}{\sqrt{2\pi|\mathbf{B}+\boldsymbol{\Sigma}|}}\exp\Big(-\frac{(\mathbf{v}^1-\mu)^T(\mathbf{B}+\boldsymbol{\Sigma})^{-1}(\mathbf{v}^1-\mu)}{2}\Big)}$$
$$= \frac{1}{N^1}\sqrt{\frac{|\mathbf{B}+\boldsymbol{\Sigma}|}{|\boldsymbol{\Sigma}|}} \exp\Big(-\frac{(\mathbf{v}^1-\mu)^T\big(\boldsymbol{\Sigma}^{-1}-(\mathbf{B}+\boldsymbol{\Sigma})^{-1}\big)(\mathbf{v}^1-\mu)}{2}\Big) \tag{44}$$

Finally, the closed form of $\hat{f}(\mathbf{v^1})$ is derived as follows.

$$
\begin{aligned}
\hat{f}(\mathbf{v}^1) &= \frac{1}{N^1} \sum_{i=1}^{N^1} K(\mathbf{v}^1 - \mathbf{v}_i^1) \sqrt{\frac{|\mathbf{B} + \boldsymbol{\Sigma}|}{|\boldsymbol{\Sigma}|}} \exp\Big( - \frac{(\mathbf{v}^1 - \mu)^T (\boldsymbol{\Sigma}^{-1} - (\mathbf{B} + \boldsymbol{\Sigma})^{-1})(\mathbf{v}^1 - \mu)}{2} \Big) \\
&= \frac{\sqrt{(2\pi)^{N^1 - 1}}}{N^1} \sum_{i=1}^{N^1} \prod_{j=1}^{N^1} \frac{1}{b_j} \mathcal{K}\big(\frac{\mathbf{v}_j^1 - \mathbf{v}_{ij}^1}{b_j}\big) \sqrt{\frac{|\mathbf{B} + \boldsymbol{\Sigma}|}{|\boldsymbol{\Sigma}|}} \exp\Big( - \frac{(\mathbf{v}^1 - \mu)^T (\boldsymbol{\Sigma}^{-1} - (\mathbf{B} + \boldsymbol{\Sigma})^{-1})(\mathbf{v}^1 - \mu)}{2} \Big) \\
&\propto \frac{1}{N^1} \sum_{i=1}^{N^1} \prod_{j=1}^{N^1} \frac{1}{b_j} \mathcal{K}\big(\frac{\mathbf{v}_j^1 - \mathbf{v}_{ij}^1}{b_j}\big) \sqrt{\frac{|\mathbf{B} + \boldsymbol{\Sigma}|}{|\boldsymbol{\Sigma}|}} \exp\Big( - \frac{(\mathbf{v}^1 - \mu)^T (\boldsymbol{\Sigma}^{-1} - (\mathbf{B} + \boldsymbol{\Sigma})^{-1})(\mathbf{v}^1 - \mu)}{2} \Big)
\end{aligned}
\tag{45}
$$

where $\mathcal{K}$ represents a product Gaussian kernel in Eqs.(7) and (8).

### A.7  The Derivative of the Kernel Density Estimation $f(\mathbf{v^1})$

As $\mathbf{B}$ and $\boldsymbol{\Sigma}$ are diagonal matrices, we have

$$
\sqrt{\frac{|\mathbf{B} + \boldsymbol{\Sigma}|}{|\boldsymbol{\Sigma}|}} = \sqrt{\prod_{j=1}^{N^1} \Big( 1 + \frac{(b_j)^2}{(\sigma_j)^2} \Big)}
\tag{46}
$$

and

$$
\boldsymbol{\Sigma}^{-1} - (\mathbf{B} + \boldsymbol{\Sigma})^{-1} = diag\Big( \frac{(b_1)^2}{(\sigma_1)^2((\sigma_1)^2 + (b_1)^2)}, \cdots, \frac{(b_{N^1})^2}{(\sigma_{N^1})^2((\sigma_{N^1})^2 + (b_{N^1})^2)} \Big)
\tag{47}
$$

Let $\bar{f}(\mathbf{v}^1) = \frac{1}{N^1} \sum_{i=1}^{N^1} \prod_{j=1}^{N^1} \frac{1}{b_j} \mathcal{K}\big(\frac{\mathbf{v}_j^1 - \mathbf{v}_{ij}^1}{b_j}\big)$ be the original kernel density estimator, we have

$$
\begin{aligned}
\hat{f}(\mathbf{v}^1) &= \bar{f}(\mathbf{v}^1) \sqrt{\frac{|\mathbf{B} + \boldsymbol{\Sigma}|}{|\boldsymbol{\Sigma}|}} \exp\Big( - \frac{(\mathbf{v}^1 - \mu)^T (\boldsymbol{\Sigma}^{-1} - (\mathbf{B} + \boldsymbol{\Sigma})^{-1})(\mathbf{v}^1 - \mu)}{2} \Big) \\
&= \bar{f}(\mathbf{v}^1) \sqrt{\prod_{j=1}^{N^1} \Big( 1 + \frac{(b_j)^2}{(\sigma_j)^2} \Big)} \exp \sum_{j=1}^{N^1} \Big( - \frac{(b_j)^2 (\mathbf{v}_j^1 - \mu_j)^2}{2(\sigma_j)^2((\sigma_j)^2 + (b_j)^2)} \Big)
\end{aligned}
\tag{48}
$$

Thus, its derivative is generated as follows.

$$
\begin{aligned}
\frac{\partial \hat{f}(\mathbf{v^1})}{\partial b_j} &= \bar{f}'(\mathbf{v}^1) \sqrt{\prod_{l=1}^{N^1} \Big( 1 + \frac{(b_l)^2}{(\sigma_l)^2} \Big)} \exp \sum_{l=1}^{N^1} \Big( - \frac{(b_l)^2 (\mathbf{v}_l^1 - \mu_l)^2}{2(\sigma_l)^2((\sigma_l)^2 + (b_l)^2)} \Big) + \\
&\quad \bar{f}(\mathbf{v}^1) \frac{1}{\sqrt{\prod_{l=1}^{N^1} \Big( 1 + \frac{(b_l)^2}{(\sigma_l)^2} \Big)}} \prod_{l \neq j} \Big( 1 + \frac{(b_l)^2}{(\sigma_l)^2} \Big) \frac{b_j}{(\sigma_j)^2} \exp \sum_{l=1}^{N^1} \Big( - \frac{(b_l)^2 (\mathbf{v}_l^1 - \mu_l)^2}{2(\sigma_l)^2((\sigma_l)^2 + (b_l)^2)} \Big) + \\
&\quad \bar{f}(\mathbf{v}^1) \sqrt{\prod_{l=1}^{N^1} \Big( 1 + \frac{(b_l)^2}{(\sigma_l)^2} \Big)} \exp \sum_{l=1}^{N^1} \Big( - \frac{(b_l)^2 (\mathbf{v}_l^1 - \mu_l)^2}{2(\sigma_l)^2((\sigma_l)^2 + (b_l)^2)} \Big) \frac{(\mathbf{v}_j^1 - \mu_j)^2 \cdot b_j}{2((\sigma_j)^2 + (b_j)^2)^2} \\
&= \Big( \bar{f}'(\mathbf{v}^1) + \bar{f}(\mathbf{v}^1) \frac{b_j}{(\sigma_j)^2 + (b_j)^2} + \\
&\quad \bar{f}(\mathbf{v}^1) \frac{(\mathbf{v}_j^1 - \mu_j)^2 \cdot b_j}{2((\sigma_j)^2 + (b_j)^2)^2} \Big) \sqrt{\prod_{l=1}^{N^1} \Big( 1 + \frac{(b_l)^2}{(\sigma_l)^2} \Big)} \exp \sum_{l=1}^{N^1} \Big( - \frac{(b_l)^2 (\mathbf{v}_l^1 - \mu_l)^2}{2(\sigma_l)^2((\sigma_l)^2 + (b_l)^2)} \Big)
\end{aligned}
\tag{49}
$$

## Footnotes

[1]https://en.wikipedia.org/wiki/Normality_test