[Reviews · NeurIPS 2020]

Review 1

Summary and Contributions: The author proposed a model-agnostic method for generating the graph node adversarial samples and attacking some graph matching models. Specifically, a kernel density estimation function is proposed for pushing attacked nodes (around the target node) to aggregate together. Hence, the attacked node can confuse the graph model to make a wrong decision when matching the pairwise nodes. Also, a meta learning projected gradient method is proposed to select the attack start nodes.

Strengths: The author conducts a solid analysis of their proposed methods and also evaluate their attack methods on multiple graph models. Besides, the comparison baseline models are also solid.

Weaknesses: In the experiment part, the author should show some evidence of the unnoticeable perturbations on the graph. Also, could the author shortly explain why you chose the Gaussian model on parameter estimation?

Correctness: The whole method is valid, and the author showed enough analysis on their methodology description.

Clarity: The paper is well organized and written except for some little flaws. (E.g. equation 5 is somehow unexpected, the author should better introduce the equation before the equation.)

Relation to Prior Work: The paper firstly introduces the kernel density estimation method (KDE) into graph adversarial attack task. Also, few work are conducted on adversarial attack to graph matching model, which is an important work to evaluate the robustness of the graph matching models

Reproducibility: Yes

Additional Feedback: The rebuttal indeed improved the paper, and I will increase the score.


Review 2

Summary and Contributions: The paper proposed an attack method on GNN. They utilize a kernel density estimation function to estimate the densities of nodes and generate perturbations under a specific budget by pushing attacked nodes to dense regions in two graphs. Moreover, they developed meta learning based projected gradient descent method to optimize the objective function.

Strengths: 1. The method is novel and sound. The theoretical analysis is comprehensive and convincing. 2. The meta-learning helps in finding good attack starting points that alleviate the overlarge search domain on large graphs. 2. The experimental results demonstrated the proposed method.

Weaknesses: 1. What's the time complexity of the attack method? Since the meta-learning involved in a repeatedly training/attacking procedure, I'm wondering the time cost could be very high to find the adversarial nodes especially compare with existing methods. 2. Recently, a lot of defense methods against adversarial attacks on GNN have been proposed, such as [1-4]. I think the authors should at least choose one to demonstrate attack is still useful even under such defenses. 3. The related work part is missing, so I yield a question that by leverage meta-learning to help generate adversarial attacks is shown in [5]. So the difference should be clarified. Minor question: It seems that the proposed method is getting more powerful as the dataset comes larger from Table 2. Any insight can be obtained from this? [1] Certifiable Robustness and Robust Training for Graph Convolutional Networks [2] Adversarial Examples on Graph Data: Deep Insights into Attack and Defense [3] Topology Attack and Defense for Graph Neural Networks: An Optimization Perspective [4] All You Need is Low (Rank): Defending Against Adversarial Attacks on Graphs [5] Adversarial Attacks on Graph Neural Networks via Meta Learning

Correctness: The claims are proofed in the paper and maybe correct.

Clarity: The paper is well written and many experimental details are stated in the appendix.

Relation to Prior Work: Not very comprehensive.

Reproducibility: Yes

Additional Feedback: Overall, I agree the novelty of this paper, I'd like to change my score if the authors can answer my questions in [Weaknesses] part. =================================== After read the rebutall, the author addressed my question well and I decided to rasie my score to 6. The contribution needs to be clarified in the revision.


Review 3

Summary and Contributions: In this paper, the authors propose a method for adversarial attacking graph matching methods based on a kernel density estimation approach and a meta-learning based gradient descent method. Theoretical results are provided in deriving the KDE and experiments demonstrate the effectiveness of the proposed method in successfully downgrading a range of recent deep graph matching methods. In the supplementary material, the authors also generalize their method in node classification and link prediction tasks.

Strengths: (+): The authors study a novel problem, i.e., adversarial attacks of graph matching. (+): The literature survey is quite comprehensive. (+): Experiments demonstrate the effectiveness of the proposed method. (+): The proposed method may be generalized to other settings such as node classification.

Weaknesses: (-): The proposed method does not precisely correspond to the motivation. (-): Some technical details and experimental settings are not clear. The negative points are explained more specifically as follows. (1) The authors claimed the main attack strategy as “a kernel density estimation approach to push attacked nodes to dense regions in two graphs, such that they are indistinguishable from many neighbors” and focus on deriving the KDE in Section 3. However, in the objective function Eq. 9, KDE is only adopted as a sort of regularization. I feel it is the first term, which explicitly pushes away ground-truth matching points, that really matters and it does not depend on the sophisticated KDE. The authors should prove that the KDE term is indeed helpful, e.g., by an explicit theorem or conducting an ablation study. (2) I am also not sure why the authors claimed using KDE “reduces the possibility of perturbation detection by humans or defender programs” since the level of attack, i.e., whether the permutation is perceptible or not, is only determined by the budget. (3) As for the experimental setting, it seems that most baselines are designed for GNN-based node classification tasks and how to adapt them in the graph matching problem remains unexplained (e.g., do you still use misclassifying node labels as the objective function?). In addition, following (1), I think the authors should also compare with the most intuitive method of directly maximizing the first term in Eq. 9. (4) Also, what’s the adopted projection M function? Do you use a surrogate model as [98] or do you need the actual graph matching model?

Correctness: Yes, as far as I can tell.

Clarity: Yes

Relation to Prior Work: Yes

Reproducibility: Yes

Additional Feedback: (1) I suggest using a consistent metric, i.e., either mismatching rate or precision, in all the figures, since they show explicitly the opposite trends and mixing them is confusing. (2) I also suggest the authors trying/extending their method to an “unsupervised” setting, e.g., not using ground-truth matching pairs, which will make the proposed model more practical, e.g., for a social network to anonymize, since it may not be feasible to collect such ground-truths in the first place. (3) A brief discussion of how KDE can be utilized to other graph tasks beyond attacking graph matching may also be interesting. ===Updates=== (1) I appreciate the ablation study in Figure 5 about KDE and MLPGD, but this is not what I asked. Let me rephrase to see whether I understand this correctly: the authors termed all Eq.5 as the KDE, and I am curious in this equation, whether the first part, i.e. pushing away matching nodes, or the second part, i.e., maximizing density, actually matters. I believe this is important since only the second term is the actual KDE (we can push away matching nodes even we do not know KDE). (2) I agree with the authors that “small attack budget is not enough for imperceptible attacks”. However, the examples in the rebuttal slightly improve the motivation but do not entirely clarify it since what the authors suggested seems to be a degree-related attack budget. Using the authors’ example, for a node with two edges, how KDE can make the attack imperceptible since changing one edge is inevitably obvious? The square example seems reasonable at first, but raises questions at second thoughts since this is not how humans/most machine learning models analyze graphs (since we cannot assess the graph density easily as in assessing crowd density) . More appropriate examples are needed (e.g., a synthesis graph example?). (3) Baseline setting: I appreciate the clarification and believe it’s important to add these details in an updated version. Considering that the rebuttal addressed some of my concerns (though not entirely) and this paper studies a new question, I have raised my rating to 6.


Review 4

Summary and Contributions: The paper studies the problem of attacking the graph matching models. The authors propose an adversarial attack model to perturb the structure and degrade the quality of graph matching. First, a kernel density estimation is used to maximize node densities to derive imperceptible perturbation. Then a meta-learning-based PGD method is utilized to choose the attack starting points to improve the search performance. The experimental results show the effectiveness of their approach.

Strengths: The paper is well motivated, and the idea is very interesting. The paper is the first work to design the strategy to attack the graph matching model. The theoretical analysis is detailed. The empirical evaluation setting is reasonable, the experiment is complete, and the results look great.

Weaknesses: The authors assume that the graph data follow the Gaussian distribution, but I cannot find any evidence to support the assumption. The empirical analysis or reference should be provided.

Correctness: In my opinion, the method and claims are correct. Also, the empirical methodology is correct.

Clarity: The motivation for the paper is clear. Also the motivations for using the kernel density estimation and the meta-learning-based PGD are provided. But the methodology part is not very easy to follow.

Relation to Prior Work: Yes

Reproducibility: No

Additional Feedback: ---Updates--- The response from the authors solves my concern, and my score remains the same. Please add the references and results in the paper.

[Author Response · NeurIPS 2020]

Table 1: Mismatching rate (%) with 5% perturbed edges

| Attack Model | AS | | | SNS | | | DBLP | | |
|---|---|---|---|---|---|---|---|---|---|
| | SNNA | CrossMNA | DGMC | SNNA | CrossMNA | DGMC | SNNA | CrossMNA | DGMC |
| Clean | 53.9 | 46.6 | 34.7 | 45.2 | 50.4 | 41.6 | 56.1 | 51.9 | 63.2 |
| GMA+Robust Training [7] | 62.6 | 58.5 | 53.0 | 56.2 | 66.5 | 51.8 | 71.8 | 71.0 | 77.3 |
| Vaccinated GMA [8] | 62.1 | 59.9 | 53.2 | 58.7 | 67.6 | 54.6 | 70.4 | 68.6 | 74.9 |
| GMA | **64.2** | **62.9** | **54.9** | **61.2** | **69.6** | **55.7** | **74.2** | **74.3** | **80.7** |

We would like to thank the four reviewers for the helpful and constructive comments. We have tried our best to clarify the concerns and comments by all four reviewers.

**1. Ablation study of unnoticeable perturbations and KDE (Reviewers: 1+3)**

We have included the ablation study in Figure 5 in Page 8 in the submission. Compared with our proposed GMA model with the full support of both KDE and MLPGD components, one variant, GMA-KDE, only uses our proposed KDE and density maximization to generate imperceptible attacks. GMA-KDE achieves the better attack performance than GMA-MLPGD, another version that only employs our proposed MLPGD to well choose good attack starting points. A rational guess is that it is difficult to correctly match two nodes when they lie in dense regions with many similar neighbors, although the main goal of KDE is to generate imperceptible attacks.

**2. Gaussian model on parameter estimation (Reviewers: 1+4)**

Many papers assume graph representations follow Gaussian distributions, which allows to capture graph dynamics and uncertainty [1-4]. We have run the Shapiro-Wilk test, where if the P-Value of test $> 0.05$, the data is thought to follow a Gaussian distribution. In our test, the P-Values on AS and SNS are 0.668 and 0.543, which indicate two graphs follow Gaussian distribution. We will include the results in the submission. [1] Variational Graph Auto-Encoders, NIPS 2016. [2] Adversarial Network Embedding. AAAI 2018. [3] Deep Gaussian Embedding of Graphs: Unsupervised Inductive Learning via Ranking. ICLR 2018. [4] Dynamic Embedding on Textual Networks via a Gaussian Process. AAAI 2020.

**3. Time complexity of the attack method (Reviewer 2)**

Based on [6], the complexity of meta learning is $O(d^2)$, where $d$ is the problem dimension. In our case, it is the number of nodes in each graph ($N$). Both density estimation and PGD have complexity of $O(N^2)$. Thus, the overall complexity is $O(N^2)$, which is the same as most existing attack methods that search the entire graphs to find the weakest edges to attack. [6] On the Convergence Theory of Gradient-Based Model-Agnostic Meta-Learning Algorithms. AISTATS 2020.

**4. Validate attack performance under defenses (Reviewer 2)**

We test two recent defense methods on generated attacks by our GMA model: [7] uses min-max adversarial training for defense and [8] vaccinates attack with low-rank approximations. As shown in Table 1, even with the defense, our GMA model can still achieve very high mismatching rate. We will include these results in the submission. [7] Topology Attack and Defense for Graph Neural Networks: An Optimization Perspective, IJCAI 2019. [8] All You Need is Low (Rank): Defending Against Adversarial Attacks on Graphs, WSDM 2020.

**5. Difference between our model and [5] that leverages meta-learning to generate attacks (Reviewer 2)**

[5] conducts adversarial attacks on global node classification of a single graph. It aims to solve a bilevel optimization problem: (1) training classification on graphs and (2) attacking graphs. It gradually improves attack performance by using meta learning to iteratively solve the above two problems. Our model utilizes meta learning to find good attack starting points in two graphs. [5] Adversarial Attacks on Graph Neural Networks via Meta Learning, ICLR 2019.

**6. The motivation and transferability of imperceptible attacks generated by KDE (Reviewer 3)**

To our best knowledge, our work is the first to integrate small attack budget and density estimation and maximization to produce imperceptible attacks. Real-world graphs often have imbalanced node degree distribution, i.e., some nodes have low degree but some have high degree. Only considering the budget is not enough. For example, removing 3 edges from a high-degree node with 20 edges is imperceptible. However, for a low-degree node with only 2 edges, even if removing only 1 edge, the change is obvious. On the other hand, from the viewpoint of density, if 100 people join a square with 10K people, the change is unnoticeable. But even if 1 person enters an empty square, the change is obvious. The above motivation is based on the intrinsic characteristics of real graphs and is irrelevant to any graph learning tasks. Thus, it is naturally transferred to other graph applications. Moreover, we have evaluated the attack performance of GMA on two applications of node classification network embedding in pp. 5-6 in the supplementary materials.

**7. Baseline selection in the experiments (Reviewer 3)**

The majority of existing efforts focus on adversarial attacks on single graph learning, especially node classification. To our best knowledge, this work is the first to study adversarial attacks on graph matching. There are no other graph matching baselines available. In fact, we have replaced the original losses in the baselines with the matching loss for fair comparison in our current experiments in the submission.

[Meta-Review · NeurIPS 2020]

The reviewers reached a concensus of accepting the paper. The authors propose novel and sound methods, supported with appropriate theoretical analysis, to address the problem of attacking graph matching, which is considered by the reviewers as a new problem. The reviewers are satisfied with the settings of empirical evaluation and experimental results. The authors are encouraged to carefully revise their paper to address some remaining concerns of the reviewers.